# Wearable bio-adhesive metal detector array (BioMDA) for spinal implants

Jian Li [1,2,12], Shengxin Jia [1,2,12], Dengfeng Li [1,2] ✉, Lung Chow[1], Qiang Zhang [1], Yiyuan Yang[3], Xiao Bai[1], Qingao Qu[1], Yuyu Gao[1], Zhiyuan Li[1], Zongze Li[1], Rui Shi[1], Binbin Zhang[1,2], Ya Huang [1,2], Xinyu Pan[1], Yue Hu[1], Zhan Gao[1], Jingkun Zhou [1,2], WooYoung Park [1], Xingcan Huang [1], Hongwei Chu[1], Zhenlin Chen [1,2], Hu Li[1], Pengcheng Wu [1], Guangyao Zhao[1], Kuanming Yao [1], Muhamed Hadzipasic[4], Joshua D. Bernstock[5,6], Ganesh M. Shankar [4] ✉, Kewang Nan [7,8] ✉, Xinge Yu [1,2,9] ✉ & Giovanni Traverso [3,10,11] ✉

Dynamic tracking of spinal instrumentation could facilitate real-time evaluation of hardware integrity and in so doing alert patients/clinicians of potential failure(s). Critically, no method yet exists to continually monitor the integrity of spinal hardware and by proxy the process of spinal arthrodesis; as such hardware failures are often not appreciated until clinical symptoms manifest. Accordingly, herein, we report on the development and engineering of a bio-adhesive metal detector array (BioMDA), a potential wearable solution for real-time, non-invasive positional analyses of osseous implants within the spine. The electromagnetic coupling mechanism and intimate interfacial adhesion enable the precise sensing of the metallic implants position without the use of radiation. The customized decoupling models developed facilitate the precise determination of the horizontal and vertical positions of the implants with incredible levels of accuracy (e.g., <0.5 mm). These data support the potential use of BioMDA in real-time/dynamic postoperative monitoring of spinal implants.

Spinal hardware has proved crucial in treating spinal deformities, injuries, degenerative diseases, and other conditions that affect both spinal integrity and function. The use of spinal hardware allows for the correction of spinal alignment, relief from pain, and/or prevention of further degenerative processes[1–3]

Of note, spinal fusion places incredible demands on implanted hardware due in part to dynamic loading force(s), limited thickness of the spinal pedicles, and often suboptimal bone quality/density[4,5]. Consequently, implants used in spinal fusion carry a higher risk of postoperative failure as compared to those used in other orthopedic

[1]Department of Biomedical Engineering, City University of Hong Kong, Hong Kong, China. [2]Hong Kong Centre for Cerebro-Cardiovascular Health Engineering (COCHE), Hong Kong, China. [3]Department of Mechanical Engineering, Massachusetts Institute of Technology, Cambridge, MA, USA. [4]Department of Neurosurgery, Massachusetts General Hospital, Harvard Medical School Boston, Massachusetts, USA. [5]Department of Neurosurgery, Brigham and Women's Hospital, Harvard Medical School, Boston, MA, USA. [6]David H. Koch Institute for Integrative Cancer Research, Massachusetts Institute of Technology, Cambridge, MA, USA. [7]College of Pharmaceutical Sciences, Zhejiang University, Hangzhou, China. [8]Department of Gastroenterology Surgery, The Second Affiliated Hospital, Zhejiang University School of Medicine, Hangzhou, Zhejiang 310000, China. [9]City University of Hong Kong Shenzhen Research Institute, Shenzhen 518057, China. [10]Division of Gastroenterology, Hepatology and Endoscopy, Brigham and Women's Hospital, Harvard Medical School, Boston, MA, USA. [11]Broad Institute of MIT and Harvard, Cambridge, MA, USA. [12]These authors contributed equally: Jian Li, Shengxin Jia. ✉e-mail: dengfli2-c@my.cityu.edu.hk; gshankar@mgh.harvard.edu; knan@zju.edu.cn; xingeyu@cityu.edu.hk; cgt20@mit.edu

surgeries[6,7]. Critically, understanding the position of implants during spinal fusion can also help assess the progress of bone fusion (i.e., arthrodesis) and in so doing evaluate the clinical effectiveness of the surgery.

In current practice, radiographic imaging techniques such as computed tomography (CT) are commonly used to assess the integrity/position of spinal implants/constructs should any clinical concerns manifest. However, such imaging modalities are expensive and expose patients to relatively high doses of radiation[8]. Alternative invasive solutions have been proposed for the long-term monitoring of spinal fusion via a self-powered synchronized dynamical system[9], yet enthusiasm for such an approach has been limited given the inherent risk of infection, added inflammation, and possible perturbations to the fusion process itself.

In comparison, wearable technologies offer a promising alternative and may be employed for non-invasive, real-time monitoring of a litany of clinically relevant conditions, ranging from the measurement of physical indexes to the assessment of tissue status[10–12]. In line with this, emerging wearable imaging solutions, such as ultrasound[13,14] and electromagnetic-based imaging[15–17], have been developed for real-time imaging of internal structures/organs. However, they suffer from either poor image quality due to severe interference (i.e., at skin/device and bone/implants interfaces) or an absence of decoupling models to convert response signals into positional information respectively. Additionally, high frequency wave absorption and the generation of eddy currents may lead to heat accumulation within implants[18,19], potentially causing injury to tissue[20] and in so doing perturb the fusion process.

Here we present a set of materials, sensors, and a theoretical modeling approach for a bio-adhesive stainless-steel metal implant detector array (BioMDA) in an effort to develop a non-invasive, real-time monitoring system for spinal implants. It is prudent to note that the employment of magnetostatic interaction eliminates the heat accumulation issue noted above within the implants. By engineering the biocompatible adhesive hydrogel interface layer and adopting conformal designs in BioMDA, effective and reliable sensing capabilities based on inductive coupling were in fact achieved. Using customized decoupling models, the actual position of the metal implants can be precisely tracked.

The advancements represented by the BioMDA system hold promise for a myriad of applications within medicine/surgery; particularly in the postoperative monitoring of metallic implants (i.e., throughout orthopedics). Such an approach would allow for patients to be monitored in real-time and at home, which in turn may promote decentralized health care, patient adherence/rehabilitation, and in so doing optimize post-surgical outcomes.

## Results
### Concept and working principle(s) of the BioMDA
The BioMDA system is comprised of an array consisting of 16 sensors capable of metal implant detection, a thin layer of biocompatible adhesive hydrogel which acts as an interface bonding layer, and customized theoretical decoupling model(s) capable of calculating the positional information from the measured inductive signals (Fig. 1a). With the assistance of the biocompatible adhesive layer, the sensor array can be easily and securely attached to the back of a patient's neck like a medical tape, enabling the zero power consumption, non-contact sensing of cervical pedicle screws (CPS) through the inductive coupling between the permanent magnets and the metal implants. Specifically, regular spine movement results in the variation in relative position between CPS and the BioMDA, contributing to a regular movement of magnet and the generation of inductive signals within the coil (Fig. 1a, inset). Ultimately, the noncontact sensing mechanism of BioMDA would permit the noninvasive positioning of a diverse set of orthopedic implants beyond those in the spine (e.g., KineSpring for

knee osteoarthritis, ankle implants etc) (Supplementary Fig. 1) by optimizing sensors dimension and configuration.

The sensor array referenced above has a 4 × 4 layout with 20 mm interspacing that connected via ultrathin copper/polyimide (Cu/PI, 18/12 μm in thickness) traces to ensure mechanical flexibility and structural stability. Moreover, a thin layer of hydrogel (~500 μm in thickness) was engineered with stable covalent bonds formed with both the skin and the silicone encapsulation to facilitate tight-fitting of BioMDA (Fig. 1b). This adhesive layer helped to maintain accurate relative positions between the sensors and implants to achieve stable inductive signal acquisition. Advanced laser cutting and transfer printing technologies allowed the precise fabrication of the flexible and stretchable conductive interconnections on a thin elastomer substrate layer (200 μm) that connects all sensors to the integrated ports while maintaining good mechanical robustness when undergoing up to 30% stretching and 90° twisting (Supplementary Fig. 2). An isolation film (PI, 5 μm in thickness) was involved to isolate sensors from top silicone encapsulation and provide adequate movement space for the magnets. Bilayers of polydimethylsiloxane (PDMS) were adopted as encapsulations to improve the stability and durability of the sensor array. The robust interface connection, flexible designs, and integration strategies employed in the BioMDA resulted in and ultralight system (weighting only 40.3 g) with exceptional flexibility and interface stability of the sensor array, facilitating intimate contact between the BioMDA and patient skin as well as long-term comfortability (Fig. 1c, Supplementary Fig. 3).

In clinical applications, such as postoperative tracking of spinal (cervical) fusion surgery, the BioMDA is mounted on the patient's back (neck). A series of designated movements are then performed to induce different spinal curvatures, causing tissue compression and thus changes in the relative position between the sensors and implants (Fig. 1d, Supplementary Fig. 4). These movements enable the effective capture of inducing signals from the electromagnetic coils embedded in the sensor array (Supplementary Movie 1). To realize regular monitoring of implant positions in an effort to determine whether the implants/construct have failed, e.g. CPS fracture or rod fracture (Fig. 1e, inset), the captured multichannel sensing signals are subsequently extracted and analyzed to determine the response units within the sensor array. The response signals in determined units are then passed to the decoupling models (Fig. 1e (i-iii)), where received-signal-strength (RSS)-based method is utilized to determine the horizontal distribution of the implants while a customized electromagnetic-kinematic decoupling model was adopted to calculate the vertical distance of the implants. By incorporating these models, the actual position of the metal implants can be effectively ascertained and compared with retrospective sensor data to evaluate the states of metal implants during patient recovery/rehabilitation thereby providing insights in the integrity of the hardware and by proxy osseous arthrodesis.

### Design, optimization, and characterization of the BioMDA sensing unit
The single sensing unit which serves as a core component of the BioMDA system comprises a copper toroidal coil, a silicone ring (PDMS) with an inner and outer diameter of 12 mm and 18 mm, respectively, a permanent magnet that is 8 mm in diameter and 1.5 mm in thickness, and a polyethylene terephthalate (PET) holding film (Fig. 2a, Supplementary Fig. 3a). The ring coil which is 18 mm in outer diameter, 2 mm in inner diameter, and 1 mm in thickness generates inductive electrical signals in response to the movement of the permanent magnet due to electromagnetic induction. The silicone ring, together with the PET holding film, provides initial support to the magnet while allowing sufficient movement space for it in response to external electromagnetic force(s). Figure 2b illustrates the force state of the magnet, where the combined electromagnetic and gravitational

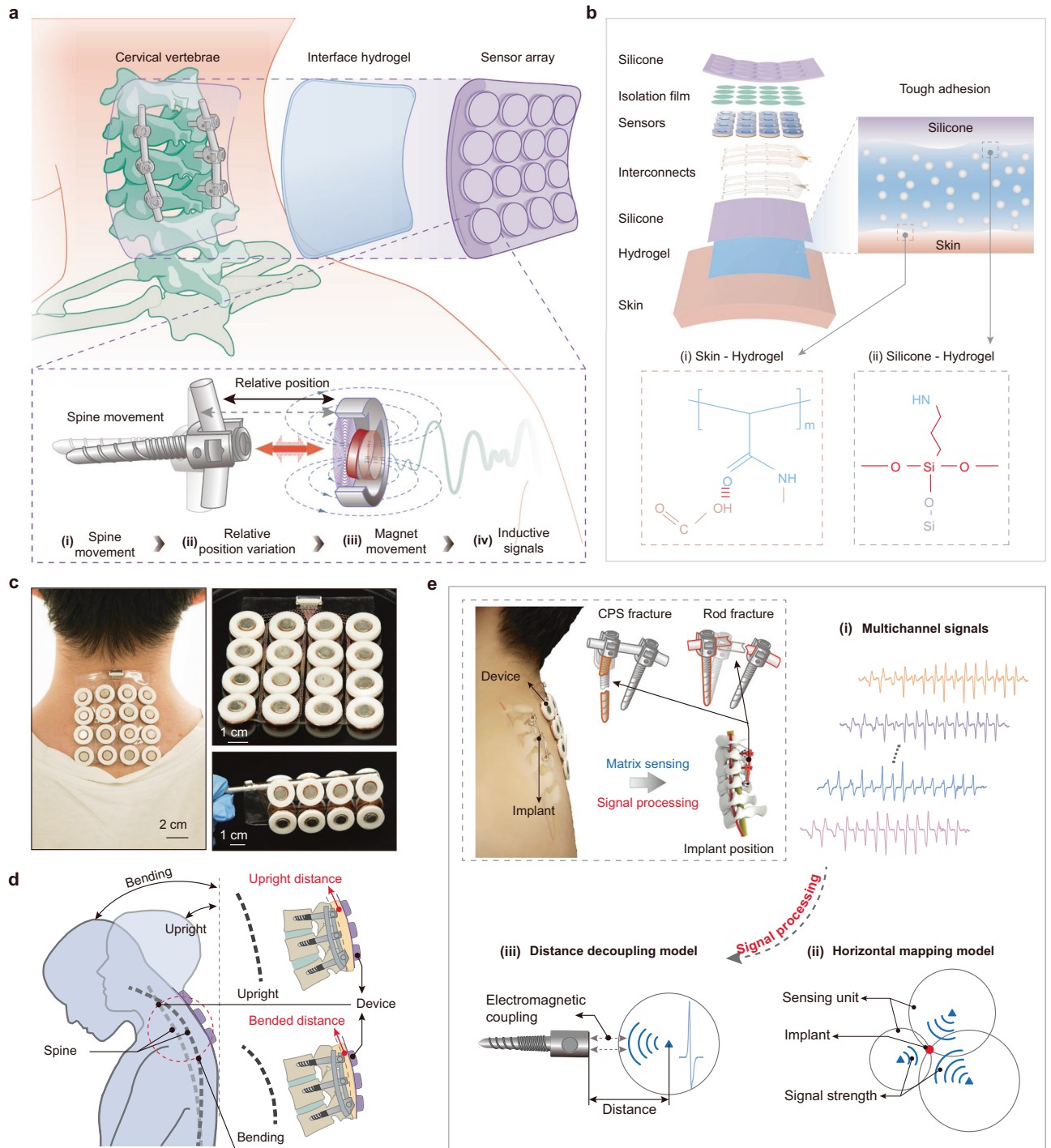

**Fig. 1 | Design and working principles of the bio-adhesive metal implant detector array (BioMDA). a** Schematic illustration of the working principle of the BioMDA. The sensor array is mounted to the skin above the cervical vertebrae, allowing for relative position changes during a set of neck movements. **b** Layered schematic illustration of the components of the BioMDA and the robust covalent connection achieved through the bio-adhesive with both silicone encapsulation and the skin. **c** Optical images showing the BioMDA mounted on user's neck (left) as well as its remarkable flexibility (right). **d** Schematic illustration showing how relative position change between the BioMDA and cervical implants are achieved via a set of bending movements. **e** Workflow diagram showing the BioMDA in diagnosing cervical pedicle screws (CPS) fracture or rod fracture through determining position changes of implants, where multichannel sensing signals arising from BioMDA are transmitted to the horizontal mapping model and the distance decoupling model sequentially for real-time 3D localization.

forces cause a bending deformation in the holding film, which, in turn, generates shear and bonding forces that counterbalance the electromagnetic force and gravity. Structural parameters such as the thickness and the central angle of the sector connection area of the PET film dictate the movement characteristics of the magnet in response to external force, thereby influencing the system's sensitivity and stability.

Next, we used a bench setup to evaluate the effects of external attractive forces on the movement of the magnet. Figure 2c presents a schematic illustration measuring the interaction force between the

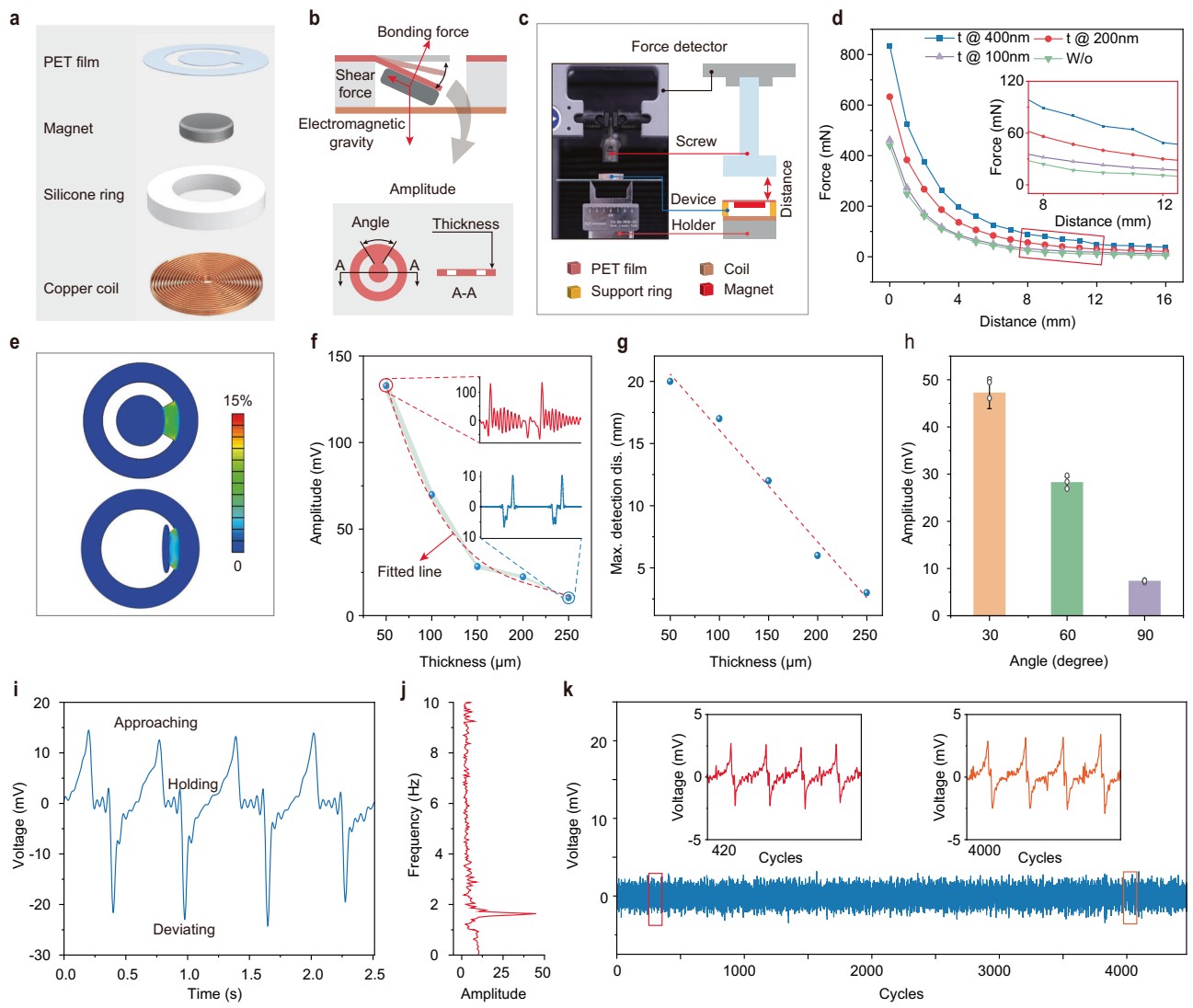

**Fig. 2 | Structural parameters optimization and characterization of the single sensing unit. a** Schematic illustration of the components comprising a single sensing unit. PET stands for polyethylene terephthalate. **b** Force state analysis of the permanent magnet in response to external attractive force and structural parameters that determine the sensing capability and stability of the unit. Schematic illustration of the platform (**c**) utilized to study the force changes with increased deviating distance (**d**) with nickel coating thickness ranging from 0 to 200 nm. **e** Finite element analysis results showing the strain distribution of the supporting PET film with thickness of 50 µm and 250 µm in response to the external attractive force of 100 mN. Measured response signal amplitude (**f**) and maximum detection distance (**g**) variation with a set of film thicknesses ranging from 50 µm to 250 µm. **h** Measured response signal amplitude with central angle of the PET film ranging from 30° to 90°, bar height, mean; error bars, s.d.; $n = 3$ independent tests. **i** Measured response signal showing the cyclic variation during implant approaching, holding, and deviating process. **j** Fast Fourier Transform (FFT) of the response signal in (**i**) showing the movement frequency of the implant. **k** Response signals in over 4000 approaching and deviating cycles showing the stability of the single sensing unit.

magnet and an exposed CPS at varying distances. The corresponding force-distance relationship is depicted in Fig. 2d, where the interactive force exhibits a quadratic decreasing trend with increased distance, consistent with the known distribution of spatial magnetic fields[21]. It should be noted that in this work, unmodified stainless-steel metal implants were employed for all experiments. For implants with lower magnetization capability, such as titanium[22,23] or zinc[24] alloys, a layer of magnetic metal can be coated on them to enhance the magnetization capability of implants to increase the interactive force and the distance limitation. To prove this, a thin layer of nickel with a thickness ranging from 100 nm to 400 nm was coated onto the stainless-steel CPS to enhance its magnetic response. Figure 2d inset highlights the increased interactive force resulting from nickel coating, particularly in the distance range of 8 mm to 12 mm, demonstrating over 100% force enhancement with 400 nm nickel coating as compared to implants without coating.

As previously mentioned, the PET holding film provides initial support to the magnet while allowing it to swing in response to external electromagnetic interactions. The mechanical properties of the holding film will influence the kinematic response and, consequently, the electromagnetic response in the coil. For instance, if the flexural stiffness of the film is too large, it will impede or weaken the motion of the magnet when subjected to a small attractive force, leading to a weak response signal in the coil, therefore greatly increasing the response threshold. On the other hand, small flexural stiffness of the film contributes to a low response threshold, thereby increasing the signal fluctuation(s) and instability of the whole system. Figure 2e illustrates the results of finite element mechanical simulation showing the strain distribution on the PET film with a thickness of 50 µm and 250 µm under the external loading of 100 mN, which is close to the loading within a detection range of 8 mm to 12 mm (Fig. 2d). The results in Fig. 2e show that the 50 µm PET swings

almost 90 degrees under the external loading, indicating poor stability of the sensor. Significant shaking of the response signal after the response peak was also experimentally observed (Fig. 2f inset) despite a larger peak voltage being generated as compared to sensors with thicker holding films. Additionally, although a thicker film contributes to stable response signals (Fig. 2f inset), a dramatic reduction in signal amplitude was observed with 250 μm thickness PET film.

To achieve a balance between the detection limit and system stability, we conducted experiments to determine the optimal combination of structural parameters including film thickness and central angle that provide a sufficient detection distance covering the normal distance range from 5 mm to 20 mm considering the distance from skin to epidural space[25,26] and the dimensions of CPS in cervical vertebrae segments C1-C7 for patients undergoing cervical spinal fusion surgery. Figure 2g, h illustrate the maximum measurement distance and response signal amplitude of the sensors with different film thicknesses, from which films with a thickness of 150 μm and a central angle of 60 degrees were shown to exhibit both satisfactory measurement limits and signal quality. Figure 2i presents the response signal in the coil during an approaching and separating cycle between a stainless-steel CPS and the sensor. A positive response peak was generated in response to the approaching of the CPS in the coil, while minor fluctuations after the response peak were observed due to the attenuated motion of the magnet under the influence of inertia and damping effect of the holding film. Subsequently, the deviation of the screw resulted in a negative response peak in the response signal due to the reverse motion of the magnet. The spectrogram of the response signal indicates the motion frequency between the sensor and the screw (Fig. 2j). Figure 2k presents the response signal over a substantial number of approach-separation cycles between the CPS and the sensor, where no significant signal fluctuation in either the waveform or amplitude was observed over 4000 cycles, indicating excellent mechanical robustness of the sensor. Additionally, temperature tracking over 1000 cycles revealed that the sensing unit experienced minimal temperature fluctuations, with less than a 2°C deviation from the minimum environmental temperature (Supplementary Fig. 5, Movie 2), ensuring long-term user safety and comfort. Moreover, dynamic tests with a CPS moving at constant distance from the sensor array at 10 mm in patterns resembling that of CITYU demonstrated the good uniformity of each sensor node with minimal signal crosstalk (Supplementary Fig. 6).

In practical applications, the environmental settings and patient conditions are often more complex than controlled lab environments. One of the least controllable variables in clinical settings is the body orientation of patients, which can introduce significant signal interference to the sensing units and decrease measurement/diagnostic accuracy. Therefore, we evaluated the signal stability across a range of device orientations, from −60° to 60° relative to the vertical plane (Supplementary Fig. 7a). The response signals in these different orientations showed no significant difference (Supplementary Fig. 7b). Additionally, we tested the device's anti-interference capability against common electromagnetic sources encountered in daily life, including permanent magnets, metal board and WIFI router. A fixed separation distance of 5 cm between the interference sources and the device was maintained to evaluate its anti-interference performance (Supplementary Fig. 8). A statical comparison on signal amplitudes revealed only minor fluctuations caused by different external interferences (Supplementary Fig. 8f), demonstrating the potential of BioMDA for real world applications. However, further reducing the separation distance between the stainless-steel plane and the device to 1 cm caused significant signal distortion (Supplementary Fig. 9), indicating that the user should avoid positioning the device too close (less than 2 cm) to the interference sources during actual use to ensure measurement accuracy.

## BioMDA interface design and characterization

The dramatic decrease in the magnetic field strength with distance poses a challenge in maintaining a stable response signal for accuracy measurements. Even a slight relative position change between the BioMDA and the CPS can result in severe fluctuations in the response signal, thus affecting the overall measurement accuracy. To address this, we developed a biocompatible, tissue-adhesive hydrogel to enable bidirectional adhesion between skin and the silicone encapsulation of the BioMDA, which cannot be achieved effectively using a traditional silicone elastomer and/or commercial adhesives. The adhesive hydrogel consists of a poly (acrylic acid) (PAA) network crosslinked with biodegradable gelatin methacrylate, along with a biodegradable gelatin network (Fig. 3a), showing strong adhesion to diverse materials (Supplementary Fig. 10) and superior biocompatibility, without skin irritation caused on 2 volunteers after 12 h continuous covering (Supplementary Fig. 11). When applied to skin, the carboxylic acid groups in PAA establish initial rapid adhesion through intermolecular bonds (e.g., hydrogen bonds and electrostatic interactions) with the skin under gentle pressures of <5 kPa in <10 s. Subsequently, the N-hydroxysuccinimide ester grafted PAA (PAA-NHS ester) forms covalent connections with the amine groups on the skin via amide reaction in the next ~10 min (Fig. 3b). To establish stable covalent connections between the adhesive and the silicone encapsulation of the BioMDA, we grafted amino groups on the surface of silicone elastomer with 3-aminopropyltriethoxysilane (APTES, see methods for details, Supplementary Fig. 12) so that covalent bonds between the hydrogel and the silicone can form through amide reactions to enable robust interface connection (Fig. 3c).

The bio-adhesive in this work exhibited a variable elastic modulus ranging from 13 to 46 kPa as the NHS-ester content varies (Supplementary Fig. 13), enabling sufficient bonding strength between the sensor array and the skin while also ensuring mechanical compatibility to maximize user comfort during the wearing process. Furthermore, the high resilience of the bio-adhesive allowed it to stretch to more than 8 times of its original length (Supplementary Fig. 14), along with its excellent reproducibility across multiple loading and unloading cycles, indicating the uniform stress distribution within its network[27] (Supplementary Fig. 14b). To assess the adhesion strength of the bio-adhesive, we conducted peeling tests and tap-shear tests to evaluate the interfacial toughness and shear strength, respectively. Silicone encapsulation material PDMS and porcine skin were chosen as interface materials for their wide applicability and close resemblance to human skin[28,29]. Figure 3d shows the results of interfacial toughness during the 180-degree peeling test for the two interface materials combinations, where over 1000 J/m² interfacial toughness is achieved in skin-to-skin interface. Notably, strong adhesion was achieved in both skin-to-skin and skin-to-PDMS, with a shear strength of 36 kPa and 12.5 kPa, respectively (Fig. 3e). Both the interface toughness and shear strength in the skin-to-skin group were higher than those in the skin-to-PDMS group due to limited hydrogen bonding (Fig. 3b, c). The interface toughness and shear strength in the PDMS-to-PDMS group presented similar values to those in the skin-to-PDMS group (Supplementary Fig. 15), further conforming that the limited hydrogen bond leads to the weaker interface adhesion performance on PDMS. It is worth noting that although the limited hydrogen bonds decrease the interface adhesion performance on PDMS, it is still sufficient for the BioMDA device used in this work. The interfacial bonding strength also exhibited good robustness, with no evident failure observed after subjecting it to 50 tensile cycles at a 20% tensile strain (Supplementary Fig. 16) and more than 5 peel-off cycles from the skin (Supplementary Fig. 17). Furthermore, the hydrogel adhesive maintained a stable connection for over 24 h at room temperature and over 3 weeks in a freezing environment at −20 °C (Supplementary Fig. 18). Additionally, the intimate interface between device and skin effectively encapsulated the bio-adhesive layer, preventing interfacial adhesion failure

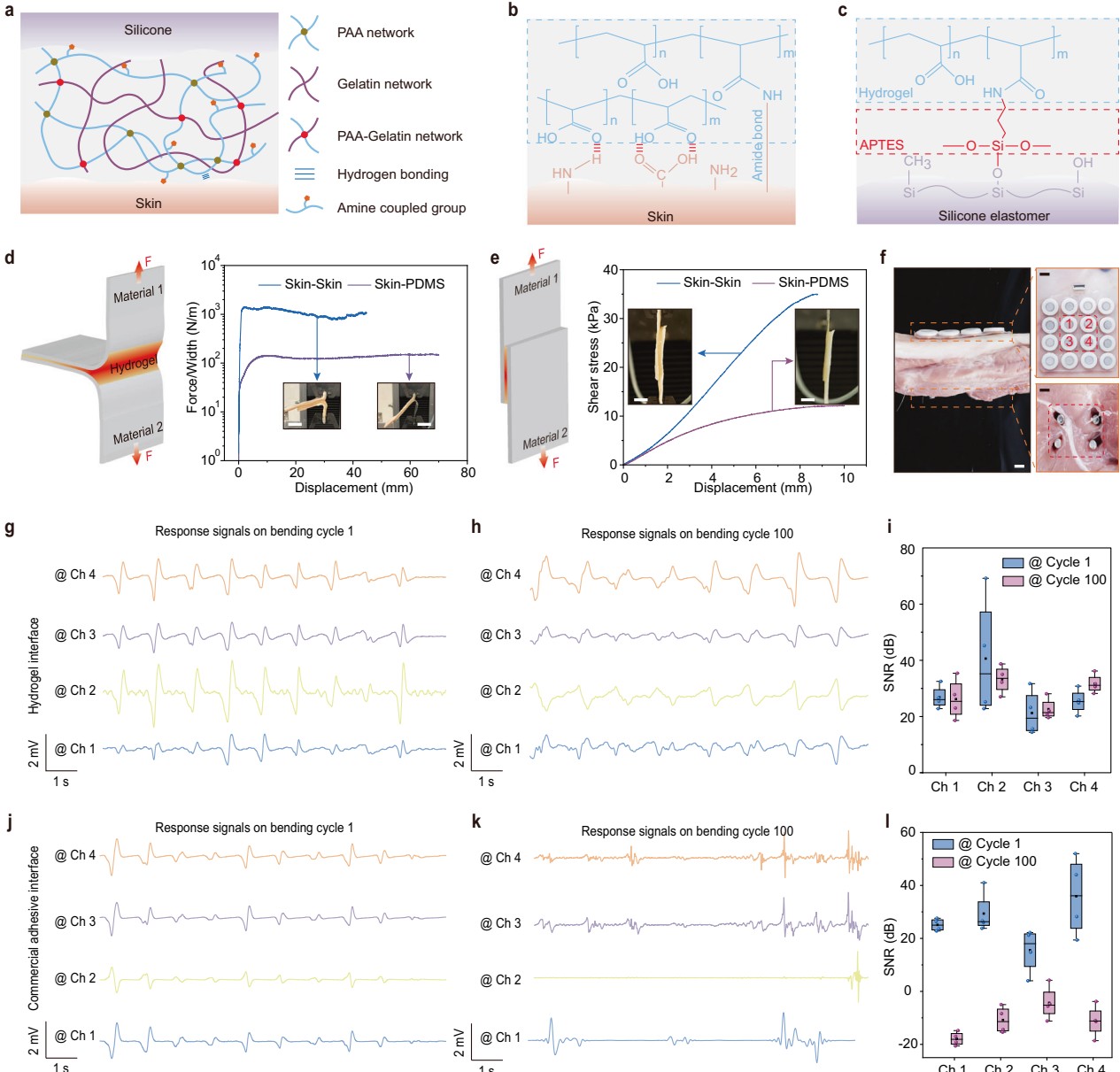

**Fig. 3 | Interfacial robustness and electrical performance evaluation.**
**a** Schematic illustration of the interfacial bonding achieved by the bio-adhesive with both the skin and the silicone encapsulation. PAA stands for poly (acrylic acid). Formation of the covalent connection between bio-adhesive and the skin (**b**) as well as the bio-adhesive and the amino grafted silicone elastomer (**c**). **d** Schematics and force variation to increased separation distance results showing the interface toughness between skin-skin and skin- polydimethylsiloxane (PDMS) connection during 180° peel test. Inset scale bars: 1 cm. **e** Schematics and shear stress variation with increased separation distance during shear test. Inset scale bars: 1 cm. **f** Optical images showing the settings of the BioMDA and metal screws in electrical performance validation experiments. Scale bars: 1 cm. Response signals from the 4 highlighted channels at bending cycle 1 (**g**) and bending cycle 100 (**h**) and Signal-to-noise ratio (SNR) comparison (**i**) with bio-adhesive as interface material. Response signals from the 4 highlighted channels at bending cycle 1 (**j**) and bending cycle 100 (**k**) and SNR comparison (**l**) with commercial adhesive as interface material. Center line, median; box limits, upper and lower quartiles; whiskers, 1.5 × interquartile range; *n* = 4 independent signals in Figs. **i** and **l**.

due to water loss (Supplementary Fig. 19). We further employed a modular design for the bio-adhesive layer and the sensor array, allowing quick replacement of the adhesive layer while maintaining the functionality of the sensing module (Supplementary Fig. 20).

The BioMDA demonstrated improved uniformity and suppressed crosstalk with the employment of the bio-adhesive interface (Supplementary Fig. 21). To evaluate the enhancement in practical sensing performances through the incorporation of the bio-adhesive, we conducted a comparative analysis of signal quality between the signals generated using the bio-adhesive in this work and those produced with a commercial double-sided adhesive (acrylic adhesive 100MP, 3 M) as

the interface material. During the tests, the BioMDA was mounted on the top side of the porcine skin, while four CPS were embedded into the porcine tissue/bone from the bottom side with a distance of ~10 mm apart (Fig. 3f). The porcine tissue was then subjected to bending deformation to mimic positional shifts between the CPS and BioMDA that frequently occur in real patient measurements (Fig. 1d). Figure 3g, j present the response signals from the four highlighted sensing units in hydrogel and the commercial adhesive group, where consistent response signals are generated across all sensing channels. However, the response signals produced in the commercial adhesive group demonstrate both smaller amplitudes and larger variations

compared to those in hydrogel group (Fig. 3j). The reduction in amplitude can be attributed to the presence of grain stain or dust on the skin, which diminishes the bonding strength, and thereby generates gaps between the sensor array and the skin. In comparison, the response signals produced in the bio-adhesive group display regular and stable responses across all channels, indicating robust adhesion at the interface (Supplementary Fig. 22).

Additionally, after subjecting the system to 100 bending cycles, the signals in the bio-adhesive group exhibited a negligible decrease in peak amplitude along with minor fluctuations across 4 channels (Fig. 3h). On the contrary, severe fluctuations were observed across all channels in the commercial adhesive group after 100 bending cycles due to the formation of air gaps at the interface due to adhesive failure. A quantitative comparison of the signal-to-noise ratio (SNR) further reveals that the response signals in the bio-adhesive group show no significant reduction in SNR (Fig. 3i), while those in the commercial double-sided adhesive group demonstrate a dramatic decrease (Fig. 3l, Supplementary Fig. 23). In practical applications, sweat accumulation poses a significant challenge for achieving robust interface adhesion and maintaining biocompatibility. While commercial double-sided adhesive has proven to be an effective solution for the device-skin interface[30], sweat accumulation at the interface can still lead to adhesion failure and skin irritation (Supplementary Fig. 24). In contrast, the good water absorption capability of the biocompatible hydrogel prevented sweat accumulation at the interface, thereby avoiding skin irritation and adhesion failure (Supplementary Figs. 24, 25, Movie 3).

## Decoupling model construction and performance evaluation

We developed a two-step decoupling model to accurately estimate the implants' vertical distance and horizontal location, respectively. Specifically, we constructed the vertical distance decoupling model based on the theoretical electromagnetic interaction between the implants and the sensing unit of BioMDA and the kinematic analysis of the chain reaction triggered by a moving implant (Fig. 4a, Supplementary Note 1). The vertical distance decoupling model serves as the fundamental sensing model of each unit of BioMDA for sensor calibration and precision evaluation. With only six calibration factors (Eq. 6), the distance decoupling model accurately simulated the electromagnetic response from the coil corresponding to a moving implant. Moreover, the calibration process of each sensing unit could be reduced to six selected pairs of distance and velocity. Such an effortless calibration process helps maintain consistency among massive production and simplifies recalibration after long-term uses.

Specifically, the electromagnetic model (Fig. 4a) in the distance decoupling model includes three parts: the simplified distribution of magnetic field distribution $B$ (Eq. 1), the electromotive force (emf) $\varepsilon_O$ following Faraday's law of induction in vacuum (Eq. 2), and the phenomenological attenuation model $e^{\gamma z}$ as the wave propagates through the medium (Eq. 3):

$$B = \frac{\mu_0 M}{2}\left[\frac{z}{\sqrt{z^2 + a^2}} - \frac{z-h}{\sqrt{(z-h)^2 + a^2}}\right] \quad (1)$$

$$\varepsilon_0(z) = -\frac{d}{dt}\int_{\sum} B dA = -\frac{\mu_0 NMAa^2}{2}\left[(z^2 + a^2)^{-1.5} - ((z-h)^2 + a^2)^{-1.5}\right]\frac{dz}{dt} \quad (2)$$

$$\varepsilon(z) = e^{-\gamma z}\varepsilon_0(z) = e^{-\gamma z}\xi_B(z)\dot{z} \quad (3)$$

where $\mu_O$ is the permeability of vacuum ($4\pi \times 10^{-7}$ H/m), $M$ is the magnetization of the permanent magnet, $z$ presents the vertical distance to the magnet and its time derivative $\dot{z}$, $a$ and $h$ presents the

diameter, and thickness of the magnet, $A$ presents the related area of the magnet flux, $\xi_B(z)$ describes the theoretical magnet field change, and $\gamma$ represents the attenuation parameter which numerical methods could easily approximate. Equation 3 theoretically models the electromagnetic interaction into a distance-velocity space. As shown in the amplified window in Fig. 4e, emf $\varepsilon(z)$ was exponentially affected by distance when an implant approached a sensing unit, while velocity may add small fluctuations. Thus, we mainly focused on the emf $\varepsilon(z)$ amplitude to simplify the moving implant's localization process. Figure 4b illustrates the difference between the emf model without and with the attenuation model as in Eq. 2 and Eq. 3, respectively. The next step was to approximate the attenuation model, considering the kinematic model.

The kinematic model inspected the coupled relationship between the movement of the implants $(z_i)$ and the magnet $(z_m)$ and decomposed the movement of the magnet into vertical and rotational primitives (Supplementary Note 2). We then decomposed the collectible emf $\varepsilon(z_m)$ accordingly (Fig. 4c, d) and approximately projected $\varepsilon(z_m)$ to the distance-velocity space of the implants $z_i$ as:

$$SF(z,\dot{z}) = sf_1(z) + sf_2(z)(\dot{z} - (kz + b)) \quad (4)$$

$$sf_*(z) = a_1 e^{-z} + a_0 \quad (5)$$

$$\hat{\varepsilon}(z_m) \approx SF(z_i,\dot{z}_i)\xi_B(\max(z_i,z_0))\dot{z}_i \quad (6)$$

where constant values $k$, $b$, and $a_*$ in both scaling functions (Eq. 5) are estimated using regression, and $z_O$ is the local minima of $\xi_B(z)$ around 2.834 mm (Supplementary Fig. 27), which relates to the structural parameter of the sensing unit. Following the attenuation model, we introduced the scaling functions in the form of $sf_1$ to approximate the exponential curve in the distance space of $z_i$, and the results are illustrated in Fig. 4d. While in the velocity space, a spin around the distance axis was observed around 11.25 mm/s (Fig. 4d) with slopes following an exponential curve of $sf_1$ (Supplementary Fig. 28). We posit that this distortion relates to the rotational primitives when the magnet swings back and forth in response to external electromagnetic interactions, and the spinning center relates to the mechanical properties of the holding film. Here, we introduce a linear fit of $z_i$ to compensate for the discrepancy in the distortion. We examined the response of the sensing unit in BioMDA when an implant approached the sensor from 15 mm to a minimum distance of 2 − 6.5 mm with varying constant speed from 7.5 to 15 mm/s with an incrementation of 0.5 mm in both distance and velocity (Fig. 4e). The implant was controlled by a Micro-Newton tester (Instron, 5942), and each test was repeated 20 times. The amplitude of the calibrated model is visualized in Fig. 4e for distances ranging from 1.25 mm to 6.5 mm and velocities ranging from 7 mm/s to 15.5 mm/s. Compared with 1525 test samples, the error between our learned model and experiment values is $1.5 \pm 1.3$ μV (75 tests lost their first samples during data acquisition). We posit that the distance decoupling model could be applied to other electromagnetic sensing systems with suitable modifications. It is worth noting that the distance decoupling model functions as a decoder to directly determine the distance between the implants and the corresponding sensing units. Due to the negligible attenuation of magnetic field strength caused by human tissue[31,32], the decoupling model can function effectively across different body types.

While the vertical distance model processes the signal from a single unit in BioMDA, the horizontal mapping model evaluates signals from multiple units in BioMDA to localize the horizontal position of the implants. When an implant is off the central axis of a sensor unit, its movement triggers multiple sensor responses. We consider the response differences between multiple sensor units analogous to the

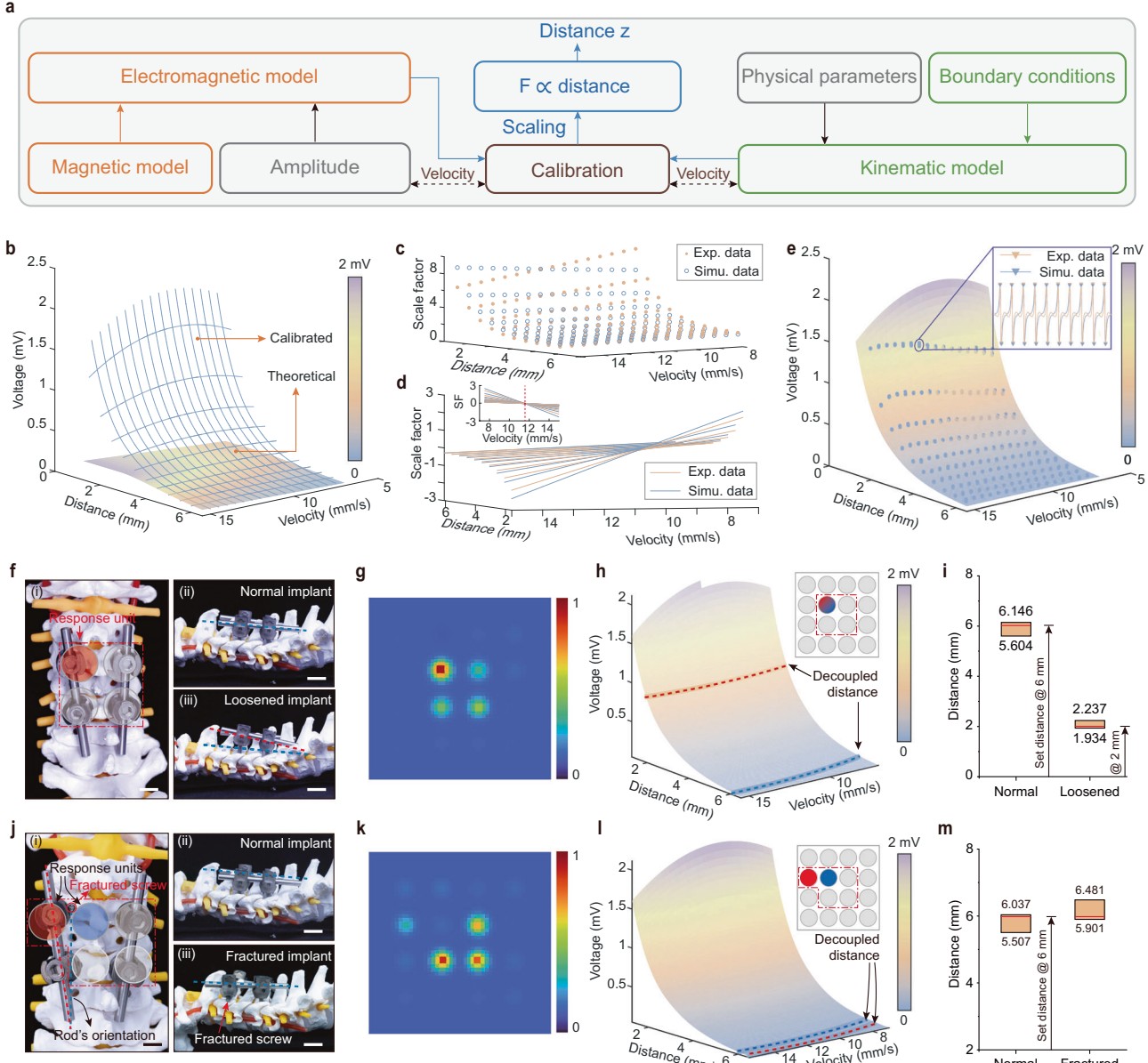

**Fig. 4 | Decoupling models construction and performance evaluation.**
**a** Framework of the electromagnetic-kinematic decoupling model to calculate the vertical distance of the implants related to BioMDA. This model calculates the vertical distance $z$ between implants and sensors by decoupling the electromagnetic and kinematic interactions, with $F$ representing the force exerted.
**b** Visualization of the difference between the theoretical model without and with the calibrated attenuation model. The calibration approximates the scaling between theoretical model $\varepsilon_O(z_i)$ to the captured inducing signal across the coil $\varepsilon(z_m)$ and can be divided into two steps scaling in the distance space (**c**) and velocity space (**d**). **e** The comparison between the calibrated model $\hat{\varepsilon}(z_m)$ and test samples is visualized for distances ranging from 1.25 to 6.5 mm and velocities ranging from 7 to 15.5 mm/sec. Optical images of screw malposition (**f**) and screw immigration (**j**) with spinal cord. **g**, **k** Decoupled spatial distribution of the implants. To estimate the vertical distance of the implants, the electromagnetic-kinematic model is sliced by the maximum amplitude of the captured inducing signal (**h**, **l**), where red lines indicate abnormal signal from malfunctioned implants and blue lines for normal implants. The localization results include an estimated range of distance (**i**, **m**). Scale bars in **f** and **j**: 1 cm.

well-suited Received Signal Strength (RSS)[33]-based localization problem based on the theoretical simulations (Supplementary Figs. 29 and 30). Specifically, we adopted Radial Basis Function kernels to generate a 2D map of received signal strength from the sensing array to determine implant malposition or migration.

With the employment of the distance decoupling model and the horizontal mapping model, we could accurately identify the common failures modes with spinal implants, such as screw loosening, screw fracture, rod fracture, and rod loosening, which collectively account for more than 90% of failures in postoperative spinal fusion processes[34,35]. In practical applications, CPS loosening or fracture

results in ineffective securing of the implant rods by the implant threads, which leads to an unstable connection between the spines and cause a decreased relative distance between CPS and the BioMDA in response to spine bending. In comparison, rod loosening, rod fracture, and CPS fracture introduce instability in the connection between the threads and the rods, leading to relative displacement between the CPS and the BioMDA.

To evaluate the sensing capability of the BioMDA, we conducted in-vitro evaluation experiments on a spinal prosthesis with two types of relative positions between implants and BioMDA, shorting in vertical distance and horizontal misalignment, included to effectively

simulate the relative position between spine and implants in normal and failure scenarios. In the case of screw loosening (Fig. 4f), we selected four CPSs with a diameter of 4.5 mm and a length of 45 mm based on the pedicle width[36]. These CPSs were inserted into the C3 and C5 segments of cervical vertebrae under the connection of two rods with a diameter of 5 mm and a length of 120 mm. To imitate the skin, a thin layer of artificial skin (PDMS, 6 mm in thickness) was employed to isolate the implants and BioMDA (Fig. 4f(ii)). The loosened screw was set to be pulled out of 4 mm, with a 2 mm vertical distance to the BioMDA (Fig. 4f(iii)). From the response unit in the BioMDA, we firstly evaluated the horizontal mapping of the implants based on the RSS-based model (Fig. 4g), where the top left response unit showed enhanced signal strength while no obvious horizontal misalignment was observed. Based on this, we analyzed response signals under two types of experimental settings (normal setting and loosened setting) based on the vertical decoupling model. Two decoupled traces were calculated, as highlighted with blue and green dotted line in Fig. 4h, representing the response signal in normal and loosened settings, respectively. Quantitative statistics on the decoupled distances presented dynamic ranges from 5.604 mm to 6.146 mm and 1.934 mm to 2.237 mm for normal implant and loosened implant, respectively (Fig. 4i). Similarly, when it comes to screw fracture failure mode, the top left screw was fractured with a misalignment of 5 mm (Fig. 4j). The horizontal mapping of the implants indicated that misalignment occurred to the top left screws with a smaller signal strength compared to those in other response units (Fig. 4k). Analysis on the two response units (highlighted in Fig. 4j) with vertical decoupling model indicated the decoupled distances (Fig. 4l), ranging from 5.507 mm to 6.037 mm and 5.901 mm to 6.481 mm for normal implants and fractured implants (Fig. 4m), respectively. Furthermore, from the vertical decoupling model, we can conclude that when the patient bends the neck to enable the effective capture of inducing signals from the BioMDA, a qualitative description of the velocity of movement, such as slow or fast, can further improve the calculation accuracy.

## Discussion

Postoperative monitoring is often crucial for the early detection and diagnosis of hardware failures within the spine (e.g., screw malposition or migration, rod fracture etc) and for the prevention of severe clinical complications such as nerve damage/paralysis, spinal instability and/or failure of fusion[37]. By implementing the BioMDA system, it may become possible to reduce patient exposure to harmful radiation whilst concurrently optimizing patient outcomes and recovery (i.e., via tracking of both the hardware construct/osseous fusion). In such a scenario, prior to abject hardware/fusion failure modifications could be put in place that might include adjuvant bracing and/or alterations in physical activity/rehabilitation.

The capabilities of BioMDA are centered on three main advances over previously reported technologies: (i) flexible design and integration whilst maintaining remarkable mechanical robustness; (ii) an engineered biocompatible adhesive which enables the creation/maintenance of intimate interface between sensors and users' skin, significantly improving the sensing capability and stability of the system; (iii) the development and optimization of theoretical decoupling models for the precise real-time localization of implants/hardware. A litany of studies including reductionist benchtop work, theoretical simulations, comprehensive experiment characterization, and validation using human phantoms have demonstrated the feasibility of all key functions/components of the BioMDA system as well as the practical utility of using it for non-contact sensing and localization of metal implants.

A number of limitations will need to be addressed in future iterations of the BioMDA system. These are centered on the materials that are able to be detected (i.e., with the understanding that medical implants have moved away from stainless steel and toward titanium)

and the depth of detection (i.e., if such a system is to be applicable in the lumbar spine). Composite materials within the final construct or the incorporation of an active magnetic field excitation may solve the former challenge (e.g., a set cap composed of a nickel alloy vs screw coating as was mentioned above). Furthermore, initial indications may focus on clinical populations that make challenges related to depth negligible (e.g., low BMI cut-offs vs pediatric hardware etc.). Further investigations will examine strategies for enhancing the sensing limits and precision for deeper implants, as well as incorporating artificial intelligence (AI)-based decoupling models for diverse types of implant detection. From a regulatory perspective it is also important to note there have been precedent FDA approvals with medical devices that contain magnets, such as magnetic devices systems for in vivo biopsies[38].

In summary, the BioMDA system represents a promising technological advancement in wearable sensing via the provision of a cost-effective real-time solution for metal implant analyses. Such promising technology will certainly be applicable beyond spinal hardware and may ultimately have a role to play throughout trauma/orthopedic surgery.

## Methods

### Fabrication of the stretchable electrodes of the BioMDA

The fabrication process started with the precise patterning (Supplementary Fig. 31) of the copper/polyimide (Cu/PI) film using an ultraviolet laser processing system (ProtoLaser U4, LPKF). Initially, a glass sheet (75 mm × 75 mm) was thoroughly cleaned using acetone, isopropyl alcohol (IPA), ethanol and deionized (DI) water to eliminate any surface residues before spin-coating a thin layer of polydimethylsiloxane (PDMS, Sylgard 184 silicone elastomer), with a cross-linker ratio of 10:1 with a thickness of 200 μm to serve as an adhesion layer. Next, a layer of Cu/PI thin film, with a thickness of 18/12 μm, was laminated onto the PDMS layer before patterning the electrode traces with the laser processing system. Water soluble tape (WST) was then employed to peel off the patterned electrode traces from the adhesive layer of PDMS, followed by cleaning the PI side with IPA to remove carbon residues during the laser patterning process. Subsequently, a layer of titanium (Ti) and silicone dioxide (SiO2) with a thickness of 5 nm and 50 nm, respectively, was deposited onto the PI side of the electrode to facilitate its robust bonding to the PDMS substrate. Both the PDMS substrate and the electrode pattern underwent oxygen plasma treatment before being bonded together under a general pressure at 80 °C in a heated oven for 30 min for the formation of covalent bonding of silicone. Finally, the WST was dissolved using DI water, and the electrode was tried before being partially encapsulated with PDMS with the assistance of sacrificial Polyvinyl alcohol (PVA) pillars to protect the welding pads. The robust covalent connection between the electrode traces and the PDMS substrate, combined with the PDMS encapsulation layer, provides remarkable mechanical robustness to the electrode, capable of undergoing over 20% stretch and over 180° twist deformation while without delamination caused (Supplementary Fig. 2). Additionally, the robust connection between the welding pads and the PDMS substrate enables stable soldering connections between the pads and sensors, significantly enhancing the mechanical robustness of the BioMDA.

### Fabrication of the single sensing unit

The single sensing unit comprises a copper coil, a silicone supporting ring, a magnet, and a polyethylene terephthalate (PET) holding film. To prepare the sensing unit, PET films with thicknesses ranging from 50 μm to 250 μm, with intervals of 50 μm, were first patterned into a concentric ring structure connected by a sector-shaped area using the laser processing system (ProtoLaser U4, LPKF). Next, the soft silicone supporting ring was fabricated through inversion molding of PDMS

(Sylgard 184 silicone elastomer, with the cross-linker ratio at 10:1) with the assistance of a 3D printed polyacrylate mold. The magnet, with a diameter of 8 mm and a thickness of 1.5 mm, was bonded to the PET holding film using a commercial double-sided adhesive (acrylic adhesive 100MP, 3 M). Finally, the copper coil, with an outer diameter of 18 mm, inner diameter of 1 mm, and a thickness of 1 mm, the soft supporting ring, and the PET film bonding with the magnet were assembled in alignment using a PDMS adhesive (Sylgard 184 silicone elastomer, with the cross-linker ratio at 30:1) to form a cohesive sensing unit.

## Preparation of the adhesive hydrogel

To prepare the adhesive hydrogel, acrylic acid, gelatin, gelatin methacrylate (GelMA, type A, gel strength 300 from porcine skin with 80% substitution), acrylic acid N-hydroxysuccinimide ester (AAC-NHS ester), and Lithium phenyl-2,4,6-trimethylbenzoylphosphinate (LAP) were obtained from Sigma-Aldrich without further purification. Next, a mixture was prepared by dissolving 30% acrylic acid, 10% gelatin, AAc-NHS ester, with concentration ranging from 0.3% to 1%, 0.1% GelMA, and 0.05% LAP in DI water. The mixture was stirred for 24 h at a temperature of 60° to obtain a homogeneous solution. The obtained solution was then filtered with 0.4 μm sterile syringe filters to remove any impurities. Subsequently, the solution was poured into glass molds covered with releasing films followed by degassing and curing in an ultraviolet light chamber (405 nm in wavelength) for 30 s for photo cross-linking.

## Integration of BioMDA

To integrate the BioMDA, the stretchable electrode, 16 sensing units, and adhesive hydrogel patch need to be first prepared using the above-mentioned methods. To facilitate the covalent bonding between the silicone encapsulation of the electrode and the bio-adhesive, (3-aminopropyl) triethoxysilane (APTES, Sigma-Aldrich) was used to graft amine functional groups onto PDMS. Specifically, the PDMS was treated with oxygen plasma (plasma power at 50 W, 20 Sccm of oxygen flow) for 10 min to activate hydroxyl groups on it (Fig. S7). Then, the activated stretchable electrode was immersed in 1% (w/w) APTES solution (1% (w/w) APTES in 50% ethanol) and incubated for 5 h at room temperature before cleaning it with isopropyl alcohol, ethanol, and DI water. Next, the prepared adhesive hydrogel film was gently pressed onto the PDMS side for over 10 s. The opposite side of the adhesive hydrogel that will be adhered to the skin was covered with a silicone oil coated releasing film (50 μm in thickness) to isolate it from dust and ease the integration of sensing units. After that, 16 sensing units were soldered to their respective electrodes to facilitate stable electrical connections. PDMS (Sylgard 184 silicone elastomer, with the cross-linker ratio at 10:1) was then used to construct a robust bond between the silicone supporting ring of the sensing units and the PDMS substrate of the electrode. An isolation layer (PI, 18 μm in diameter, 25 μm in thickness) was fabricated by a laser cutter and laminated onto the sensing units to isolate them from uncured PDMS used in the encapsulation process. Finally, PDMS was poured onto the sensing array followed by a degassing process, and cured at 80° for 2 h to form the top encapsulation layer. It is worth noting that the releasing film also acts as an isolation layer, keeping the bio-adhesive layer from dust and dehydration and can be easily peeled off before adhering the BioMDA to human skin.

## Characterization

The electrical-related data was sampled by the multichannel data acquisition system (DAQ 6510, Keithley) with a constant sampling frequency at 1000 Hz. The nickel-coated implant samples were prepared with a dual target sputtering system (Q150TS, Quorum). The thickness of coated films was measured by a stylus profilometer (DektakXT).

## Mechanical tests

Porcine skin samples were stored in phosphate buffered saline (PBS) and sealed in plastic bags. Upon tests, samples (porcine skin or aminated PDMS) were cut into 20 mm in width and 80 mm in length. To ensure the stable interface connection between tested samples, the adhesive hydrogel (20 mm in width, 50 mm in length) was adhered to samples, followed by generally pressing for 10 s. To determine the interface toughness and shear strength, the prepared samples were tested by 180-degree peel test and rap-shear test with a micro-Newton tester (5942, Instron) with a constant loading speed of 0.1 mm/s under the instruction of ASTM F2256 and ASTM F2255, respectively. The interfacial toughness was calculated by dividing two times the peak force by the width of the sample, and the shear strength was calculated by dividing the maximum force by the adhesion area.

## Mechanical simulation

Finite element analysis was utilized to model the sensing unit in studying its mechanical response to external attractive force. The simulation was conducted using commercial software ABAQUS (version 2018, Dassault System). The deformation and strain distribution of the PET supporting film with a set of thickness and central angles was studied to optimize the structural parameters of the sensing units for optimal electrical response and measurement limitations. In the simulation, the silicone elastomer PDMS was modeled with hyperelastic materials under the governing of Mooney-Rivlin energy potential model with elastic modulus (E) and Poisson's ratio at 150 kPa and 0.49, respectively. The PET supporting film was modeled using shell elements (S4R) with elastic modulus and Poisson's ratio at 2.8 GPa and 0.45, while the permanent magnet was modeled using a rigid body due to its minimal deformation during the sensing process.

## Electromagnetic simulation

Electromagnetic simulation was employed to investigate the theoretical distribution of the magnetic field and inductive coupling between the CPS and sensing units. The simulation was performed using the commercial software Maxwell (version 16.0, Ansys). The permanent magnet was modeled by a Neodymium Iron Boron magnet, with a relative permeability of 1.04 and a magnetic coercivity of $8.76 \times 10^5$ A/m. The CPS was modeled using stainless-steel, with a relative permeability of 4000 and a bulk conductivity of $1.03 \times 10^7$ S/m. To conserve computational resources while maintaining simulation accuracy, a vacuum boundary was applied with a 100 mm offset from the model boundary in x, y, and z direction. Furthermore, mesh refinement was implemented, setting a maximum element length of 0.5 mm for the CPS and 1 mm for the vacuum box, to ensure simulation accuracy.

## Ethics statement

All procedures during the BioMDA system testing from human participants are approved by Human and Artefacts Ethics Sub-Committee, City University of Hong Kong Research Committee. The informed consent of all participants was obtained prior inclusion in this study.

## Reporting summary

Further information on research design is available in the Nature Portfolio Reporting Summary linked to this article.

# Data availability

All data supporting the findings of this study are available within the article and its supplementary files. Any additional requests for information can be directed to, and will be fulfilled by, the corresponding authors. Source data are provided with this paper.

# Code availability

The code supporting the findings of this work is available at the following link: https://zenodo.org/records/12806601.

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

## Acknowledgements

This work was supported in part by: InnoHK Project on Project 2.2—AI-based 3D ultrasound imaging algorithm at Hong Kong Centre for Cerebro-Cardiovascular Health Engineering (COCHE), City University of Hong Kong (Grants No. 9667221, 924007 and 9680322), National Natural Science Foundation of China (Grants No. 62122002), Research Grants Council of the Hong Kong Special Administrative Region (Grant No. RFS2324-1S03), Shenzhen Science and Technology Innovation

Commission (Grant No. SGDX20220530111401011), the Innovation and Technology Fund of Innovation and Technology Commission (Grant No. ITS/119/22), Karl Van Tassel (1925) Career Development Professorship and. The Department of Mechanical Engineering, MIT.

## Author contributions

J.L. and S.J. contributed equally to this work. J.L, D.L., K.N., X.Y. and G.T. conceived the ideas and designed the experiments. J.L, S.J., D.L, Y.Y., M.H., J.D.B., G.M.S., K.N., X.Y. and G.T. wrote and revised the manuscript. J.L., L.C., Q.Z., X.B., Q.Q., Y.G., Z.L., Z.Z.L., R.S., B.Z., Y.H, X.P., Y.Hu., Z.G., J.Z., W.P., X.H., H.C. and Z.C., performed device design and characterization experiments and analyzed the experimental data. J.L., Q.Z., H.L., P.W. and G.Z., and K.Y., performed interface design and characterization experiments, J.L., S.J. and Q.Q., developed the decoupling models and evaluated the model accuracy. J.L. conducted the mechanical and electromagnetic simulations for structural and model optimization.

## Competing interests

The authors declare filing of a provisional patent application encompassing the work described.

J.D.B. has an equity position in Treovir Inc. and UpFront Diagnostics. J.D.B. is also a co-founder of Centile Bioscience and on the NeuroX1 scientific advisory board. The authors declare filing of a provisional patent application encompassing the work described.
