## [Peer Review File · Nature Communications]

REVIEWER COMMENTS

Reviewer #1 (Remarks to the Author):

In this paper, Jian Li et al present a wearable bio-adhesive metal detector array (BioMDA) for non-invasive continuous monitoring of the condition of spinal implants. This system can dynamically track the position and integrity of metal implants within the spine using electromagnetic coupling which does not involve radiation exposure. Authors demonstrated high accuracy in position tracking in phantom model experiments suggesting its potential utility in early detection of implant failure. The overall work is interesting and can be published after addressing the following comments and questions.

1. How does the device perform in a real-world clinical setting? As the electromagnetic interference in real world will be much stronger and more complex than lab setting. And the patient's condition can be much more complex than the phantom model.
2. What's the advantage of using bio-adhesive hydrogel in this system? As conventional double-sided tape can already stick to silicone elastomer and skin quite well. As reported in this paper (Nat Biomed Eng 4, 148–158 (2020)), a double-sided tape is enough to secure the wearable device on skin even under vigorous exercise.
3. Authors claim that the hydrogel is biocompatible. However, there is no experimental data to support this point. It's important to figure out whether skin irritation will occur, especially for patients with skin conditions.
4. How's the long-term wearability of this device? How to prevent the hydrogel from dehydration or swelling during daily activities?
5. How significant is the heating issue for this device? What temperature will the copper coil reach after long-term working?
6. What's the power consumption rate of such a device?
7. How is the effectiveness of this device in different body types? Particularly in overweight or obese patients where fat tissue might influence the signal.
8. How will the environment influence this device? Will the reading still be accurate when there is a piece of metal near the neck (this can be from a chair, bed, or decoration)?

Reviewer #2 (Remarks to the Author):

In this manuscript, Li et al. developed a wearable bio-adhesive metal detector array (BioMDA) for real-time/dynamic postoperative monitoring of spinal implants. This topic is indeed an unaddressed clinical challenge, and the proposed solution using wearable technologies is quite novel and applicable. The authors have provided systematic and solid analyses of proposed technology, including detection principles, optimizations, algorithms, and adhesive materials. There can realize real-time and continuous monitoring of metal implants at non-clinical settings. Thus, it is recommended of the manuscript be

published after the following questions:

1. Figure 3E demonstrates that the adhesion between skin and skin is stronger than the adhesion between skin and PDMS. Please provide some explanations for this observation. Additionally, is it possible to test the adhesion between PDMS and PDMS to further investigate this issue, as adhesiveness is a key factor in practical applications?
2. The authors need to clearly specify the scope of application for this device in a prominent position within the article. For instance, the authors mentioned that current device cannot be applied to non-magnetic implants. Is the device specifically designed for non-magnetic implants only?
3. What is the maximum duration the proposed adhesive be maintained on the skin? Will it come off due to sweating/movement/etc.? Can the adhesive be removed/reapplied and for how many cycles before it loses adhesion? If the adhesive is only good for single use, then the technology will be less attractive.
4. Following the previous question, there is no data of the device during actual wearing by a human, other than a few photographs in Fig. 1. The authors are suggested to do more mechanical and electrical evaluations on real human testing subjects and evaluate the device's long-term wearability, sweat-resisting ability, signal resistance to human motion artifacts etc. during actual wearing.
5. The position and data of Fig. 2I for "holding" overlap.
6. The font of the y-axis in Fig. 2D should be consistent with the other figures.
7. The authors need to double-check the mathematical symbols in lines 280-308. Some symbols appear to have inconsistent font sizes and appear compressed, such as Line 284 and Line 286.
8. This BioMDA device used self-adhesive hydrogel. What is the maximum working time of this device, and will long-term attachment to human skin cause inflammation or discomfort?
9. How to assess the comfort during the wear period of BioMDA.
10. Isopropanol (toxic) was used in the synthesis process of the material. How should one determine that the isopropanol or other harmful substances used during the material synthesis will not be harmful to the human body?

Point-by-point response NCOMMS-24-19219A

Response to comments of Referee #1

Summary comment: In this paper, Jian Li et al present a wearable bio-adhesive metal detector array (BioMDA) for non-invasive continuous monitoring of the condition of spinal implants. This system can dynamically track the position and integrity of metal implants within the spine using electromagnetic coupling which does not involve radiation exposure. Authors demonstrated high accuracy in position tracking in phantom model experiments suggesting its potential utility in early detection of implant failure. The overall work is interesting and can be published after addressing the following comments and questions.

Our response: We thank the reviewer for these positive comments. We carefully addressed these issues and revised the manuscript accordingly.

Comment 1: How does the device perform in a real-world clinical setting? As the electromagnetic interference in real world will be much stronger and more complex than lab setting. And the patient's condition can be much more complex than the phantom model.

Our response: We thank the reviewer for the valuable comment. To address concerns about the device interference, we conducted experiments involving common electromagnetic interference sources found in daily life, such as metal, magnet, and high frequency electromagnetic waves to study the influence of electromagnetic interference on the device signals and overall performance in real world conditions. **Supplementary Figs. 8a-8e** present the measured response signals under no interference and under magnets, metal board, and WIFI router interferences, respectively. The results showed no significant signal fluctuations caused by external interferences. Moreover, a statical comparison on signal amplitude exhibited nearly identical mean values and distributions (**Supplementary Fig. 8f**), with only slightly pronounced fluctuations in the WIFI group. These results indicated that the BioMDA can function effectively in real world environments with more complex electromagnetic interferences. We added an explanation to these results in the text to present the anti-interference capability of BioMDA.

For the patient's condition, body orientation and body mass index are two main concerns that may introduce measurement errors to BioMDA. To evaluate the influence of body orientations on response signals, we conducted experiments under different body orientations. **Supplementary Fig. 7** presents the responses across various device orientations, where no significant fluctuations in signal amplitude and waveforms were observed. This indicates the robust anti-interference capability of the device, enabling accurate distance detection under different body orientations.

Additionally, the distance decoupling model developed in this work utilizes the response signal to directly decode the distance between the implants and devices. Existing research has shown that human tissue may absorb electromagnetic waves, leading to signal attenuation. However, this absorption was frequency-dependent and decreases dramatically with reduced signal frequency. It has been proven that human tissue causes negligible strength attenuation to the static magnetic

field. Therefore, we can conclude that different body types will not significantly affect the strength of response signals. We added these discussions in the text to facilitate better understanding.

Modifications:

On Page 12, Line 2, we added explanations on the anti-interference capability of the device and modified the text as “In practical applications, the environmental settings and patient conditions are often more complex than controlled lab environments. One of the least controllable variables in clinical settings is the body orientation of patients, which can introduce significant signal interference to the sensing units and decrease measurement/diagnostic accuracy. Therefore, we evaluated the signal stability across a range of device orientations, from -60° to 60° relative to the vertical plane (**Supplementary Fig. 7a**). The response signals in these different orientations showed no significant difference (**Supplementary Fig. 7b**). Additionally, we tested the device’s anti-interference capability against common electromagnetic sources encountered in daily life, including permanent magnets, metal board and WIFI router. A fixed separation distance of 5 cm between the interference sources and the device was maintained to evaluate its anti-interference performance (**Supplementary Fig. 8**). A statical comparison on signal amplitudes revealed only minor fluctuations caused by different external interferences (**Supplementary Fig. 8f**), demonstrating the potential of BioMDA for real world applications. However, further reducing the separation distance between the stainless-steel plane and the device to 1 cm caused significant signal distortion (**Supplementary Fig. 9**), indicating that the user should avoid positioning the device too close (less than 2 cm) to the interference sources during actual use to ensure measurement accuracy.”

On Page 19, Line 12, we added explanation on the influence of body types on system stability and accuracy and modified the text as “It is worth noting that the distance decoupling model functions as a decoder to directly determine the distance between the implants and the corresponding sensing units. Due to the negligible attenuation of magnetic field strength caused by human tissue^{31,32}, the decoupling model can function effectively across different body types.”

Newly added references:

31. Rubio Ayala, M. *et al.* Spatiotemporal magnetic fields enhance cytosolic Ca²⁺ levels and induce actin polymerization via activation of voltage-gated sodium channels in skeletal muscle cells. *Biomaterials* **163**, 174–184 (2018).
32. Formica, D. & Silvestri, S. Biological effects of exposure to magnetic resonance imaging: an overview. *Biomed. Eng. OnLine* **3**, 11 (2004).

Supplementary Fig. 8 | Anti-interference evaluation of the device. Response signals without external interference (a), with horizontal magnet interference (b), vertical magnet interference (c), metal board interference (d), and WIFI router interference (e). f. Static comparison on signal amplitude under different interferences. Square, mean; center line, median; box limits, upper and lower quartiles; whiskers, $1.5 \times$ interquartile range; points, amplitudes in response signals; $n=40$ peak values.

Supplementary Fig. 7 | Signal comparison on different device orientation. a. Schematic illustration on device orientation and device response signals (b).

Supplementary Fig. 9 | Anti-interference capability evaluation of the device with interference source distance. a. Schematic illustration on the experimental settings between the interference source and the device. b. Measured response signals under different interference distances.

Comment 2: What’s the advantage of using bio-adhesive hydrogel in this system? As conventional double-sided tape can already stick to silicone elastomer and skin quite well. As reported in this paper (Nat Biomed Eng 4, 148–158 (2020)), a double-sided tape is enough to secure the wearable device on skin even under vigorous exercise.

Our response: We thank the reviewer for the comprehensive comment. We conducted experiments on both in vivo artificial models and actual wearing conditions to compare the adhesion performance between commercial double-sided adhesive (3M 2477p) utilized in the listed reference and the bio-adhesive in this work. The results indicated that although commercial double-sided adhesive can provide satisfied initial adhesion strength, sweat accumulation on the skin-adhesive interface posed a significant challenge on interface adhesion performance and biocompatibility. Specifically, **Supplementary Fig. 24** presents the interface anti sweat performance comparison between commercial adhesive and bio-adhesive, where interface adhesive failure occurred with commercial adhesive (**Supplementary Fig. 24a**). In contrast, no interface adhesion failure was observed in the bio-adhesive group even under a high sweat pressure (**Supplementary Fig. 24b**). Moreover, good sweat absorption capability of the bio-adhesive effectively reduced the sweat accumulation on the skin interface (**Supplementary Fig. 24c**), which significantly improve the skin irritation after long-term wearing during intense basketball training for 30 minutes (**Supplementary Fig. 25**). We added explanation on these results in the text to facilitate better understanding.

Modificatinos: On Page 16, Line 9, we added an explanation on the comparison on commercial double-sided adhesive and bio-adhesive and modified the text as “In practical applications, sweat accumulation poses a significant challenge for achieving robust interface adhesion and maintaining

biocompatibility. While commercial double-sided adhesive has proven to be an effective solution for the device-skin interface³⁰, sweat accumulation at the interface can still lead to adhesion failure and skin irritation (**Supplementary Fig. 24**). In contrast, the good water absorption capability of the biocompatible hydrogel prevented sweat accumulation at the interface, thereby avoiding skin irritation and adhesion failure (**Supplementary Figs. 24, 25, Movie 3**).

Newly added references:

30. Lee, K. et al. Mechano-acoustic sensing of physiological processes and body motions via a soft wireless device placed at the suprasternal notch. *Nat. Biomed. Eng.* 4, 148–158 (2020).

Supplementary Fig. 24 | Interface anti-sweat performance comparison between the commercial double-sided adhesive and the bio-adhesive. a. Optical images of the interface adhesion failure with the commercial double-sided adhesive due to sweat accumulation. b. Optical images of the increased surface curvature caused by internal sweat pressure in the bio-adhesive

group, proving the seamless interface between the device and the artificial skin. c. Optical images showing the sweat absorption capability of the bio-adhesive under high sweat pressure. Scales bars: 5 mm.

Supplementary Fig. 25 | Interface robustness and biocompatibility comparison between the commercial double-sided adhesive and the bio-adhesive. a. Optical images showing device-skin interface before (i) and after (ii-iv) 30 minutes of intense basketball training. Sweat accumulation and interface delamination were observed after 30 minutes of training. Additionally, skin irritation was also observed due to sweat accumulation on the skin-device interface (iv). b. Optical images showing sweat accumulation and robust device-skin interface in the bio-adhesive group. No sweat irritation was observed in the bio-adhesive group, which can be attributed to the effective sweat absorption reducing sweat accumulation. Scale bars: 1 cm.

Comment 3: Authors claim that the hydrogel is biocompatible. However, there is no experimental data to support this point. It's important to figure out whether skin irritation will occur, especially for patients with skin conditions.

Our response: We thank the reviewer for the comprehensive comment. We conducted experiments on the forearms of 2 volunteers to study the biocompatibility of proposed hydrogel. Specifically, we applied three groups of materials, including hydrogel, PDMS, and commercial double-sided adhesive, to the skin for 12 hours to assess potential skin irritation. From **Supplementary Fig. 11**, we can conclude that the hydrogel showed good biocompatibility compared to commonly used silicone elastomer PDMS and commercial double-sided adhesive. We added a description of these results to the manuscript to prove its biocompatibility.

Modificatinos: On Page 13, Line 8, we added the description on biocompatibility and modified the text as “The adhesive hydrogel consists of a poly (acrylic acid) (PAA) network crosslinked with biodegradable gelatin methacrylate, along with a biodegradable gelatin network (**Fig. 3a**), showing strong adhesion to diverse materials (**Supplementary Fig. 10**) and superior biocompatibility, without skin irritation caused on 2 volunteers after 12 h continuous covering (**Supplementary Fig. 11**)”

Supplementary Fig. 11 | Biocompatibility of the developed bio-adhesive. a-b. Optical images of two volunteer’s forearms taken before and after being covered 12 hours with hydrogel, PDMS, and commercial double-sided adhesive.

Comment 4: How’s the long-term wearability of this device? How to prevent the hydrogel from dehydration or swelling during daily activities?

Our response: We thank the reviewer for the valuable comment. Indeed, dehydration and swelling are of vital importance in maintaining the long-term stability and adhesion performance of the bio-adhesive hydrogel. We conducted experiments to study the long-term stability of the bio-adhesive hydrogel in this work and if interface delamination will be caused due to dehydration or swelling. In practical applications, the skin and silicone encapsulation of the device provides encapsulation to the bio-adhesive layer, which significantly reduces the dehydration rate. **Supplementary Fig. 19a** shows the dehydration rate comparison between fully exposed hydrogel and hydrogel in actual application scenarios (schematically illustrated in the inset figure), where encapsulated hydrogel

maintained over 65% mass after 7 days exposure in 37°C environments. In comparison, unencapsulated hydrogel lost over 60% of mass in the first 12 h. Statical comparison on relative mass (**Supplementary Fig. 19b**) further indicated the long-term stability and wearability of the hydrogel. Additionally, the adhesion performance of the bio-adhesive was evaluated under actual wearing conditions (**Supplementary Fig. 25b**), where the BioMDA was mounted on the volunteer’s back while subjecting to 30 min intense basketball training. The results showed that the swelling of sweat contributed a more biocompatible device-skin interface, without skin irritation caused, while a more robust interface adhesion compared to that in commercial double-sided adhesive group (**Supplementary Fig. 25a**). We added the explanation on these results to provide a more detailed performance evaluation on the developed bio-adhesive layer.

Modificatinos:

On Page 14, Line 23, we added the explanation on the long-term anti dehydration performance of the bio-adhesive and modified the text as “Additionally, the intimate interface between device and skin effectively encapsulated the bio-adhesive layer, preventing interfacial adhesion failure due to water loss (**Supplementary Fig. 19**). We further employed a modular design for the bio-adhesive layer and the sensor array, allowing quick replacement of the adhesive layer while maintaining the functionality of the sensing module (**Supplementary Fig. 20**).”

On Page 16, Line 9, we added the description of the long-term interface stability under actual wearing condition and modified the text as “In practical applications, sweat accumulation poses a significant challenge for achieving robust interface adhesion and maintaining biocompatibility. While commercial double-sided adhesive has proven to be an effective solution for the device-skin interface³⁰, sweat accumulation at the interface can still lead to adhesion failure and skin irritation (**Supplementary Fig. 24**). In contrast, the good water absorption capability of the biocompatible hydrogel prevented sweat accumulation at the interface, thereby avoiding skin irritation and adhesion failure (**Supplementary Figs. 24, 25, Movie 3**).”

Supplementary Fig. 19 | Anti dehydration performance of the bio-adhesive. a. Weight loss comparison on the bio-adhesive under unencapsulated state and encapsulated by device and skin layer for 7 days in 37°C environments. b. Statical comparison on the relative mass variation of two groups of hydrogel during 7 days. Points, mean; error bars, s.d.; n=3 independent samples.

Supplementary Fig. 25 | Interface robustness and biocompatibility comparison between the commercial double-sided adhesive and the bio-adhesive. a. Optical images showing device-skin interface before (i) and after (ii-iv) 30 minutes of intense basketball training. Sweat accumulation and interface delamination were observed after 30 minutes of training. Additionally, skin irritation was also observed due to sweat accumulation on the skin-device interface (iv). b. Optical images showing sweat accumulation and robust device-skin interface in the bio-adhesive group. No sweat irritation was observed in the bio-adhesive group, which can be attributed to the effective sweat absorption reducing sweat accumulation. Scale bars: 1 cm.

Comment 5: How significant is the heating issue for this device? What temperature will the copper coil reach after long-term working?

Our response: We thank the reviewer for the valuable comment. We conducted experiments to study the temperature fluctuations of single sensing unit in long-term working cycles. Specifically, we applied a cyclic approach-separation between the cervical pedicle screws (CPS) and the sensing unit with the assistance of a linear motor. Then we tracked the temperature variation of the sensing unit during over 1000 cycles (**Supplementary Movie 2**). The results presented in **Supplementary Fig. 5** indicated that only a 2.7 °C temperature increase on sensing unit was observed after 1000

cycles. Moreover, the minimum and maximum experimental temperature experienced a 3.1 °C and 3.6 °C increase, respectively (**Supplementary Fig. 5**). As a result, we can conclude that the 1000 cyclic approach-separation cycles caused a negligible temperature increase on the sensing unit.

Modificatinos: On Page 11, Line 18, we added a description on the temperature stability of the device and modified the text as “Additionally, temperature tracking over 1000 cycles revealed that the sensing unit experienced minimal temperature fluctuations, with less than a 2°C deviation from the minimum environmental temperature (**Supplementary Fig. 5, Movie 2**), ensuring long-term user safety and comfort.”

Supplementary Fig. 5 | Temperature stability of the sensing unit over 1000 cycles. a-b, Temperature and c-d, Sensing signals at cycle 1 and cycle 1000. Scale bars: 1 cm.

Comment 6: What’s the power consumption rate of such a device?

Our response: We thank the reviewer for the comment. The BioMDA utilizes the electromagnetic coupling between the permanent magnet and the implants to generate response signals in the coil and there is no power consumption during this process. We added the explanation on system power consumption in the text.

Modificatinos: On Page 6, Line 6, we modified the text as “With the assistance of the biocompatible adhesive layer, the sensor array can be easily and securely attached to the back of a patient's neck like a medical tape, enabling the zero power consumption, non-contact sensing of

cervical pedicle screws (CPS) through the inductive coupling between the permanent magnets and the metal implants.”

Comment 7: How is the effectiveness of this device in different body types? Particularly in overweight or obese patients where fat tissue might influence the signal.

Our response: We thank the reviewer for the valuable comment. Referring to the answer to **Comment 1**, we conducted experiments to study the influence of body orientation on the signal stability and found the device is capable of effectively capturing response signals across different device orientations. Moreover, existing works have shown that the statistic magnetic field show negligible attenuation to human tissue, from which we can conclude that the sensors can function well across different body types. We added explanations and discussions on these results to facilitate better understanding.

Modifications:

On Page 12, Line 2, we added explanations on the anti-interference capability of the device and modified the text as “In practical applications, the environmental settings and patient conditions are often more complex than controlled lab environments. One of the least controllable variables in clinical settings is the body orientation of patients, which can introduce significant signal interference to the sensing units and decrease measurement/diagnostic accuracy. Therefore, we evaluated the signal stability across a range of device orientations, from -60° to 60° relative to the vertical plane (**Supplementary Fig. 7a**). The response signals in these different orientations showed no significant difference (**Supplementary Fig. 7b**). Additionally, we tested the device’s anti-interference capability against common electromagnetic sources encountered in daily life, including permanent magnets, metal board and WIFI router. A fixed separation distance of 5 cm between the interference sources and the device was maintained to evaluate its anti-interference performance (**Supplementary Fig. 8**). A statical comparison on signal amplitudes revealed only minor fluctuations caused by different external interferences (**Supplementary Fig. 8f**), demonstrating the potential of BioMDA for real world applications. However, further reducing the separation distance between the stainless-steel plane and the device to 1 cm caused significant signal distortion (**Supplementary Fig. 9**), indicating that the user should avoid positioning the device too close (less than 2 cm) to the interference sources during actual use to ensure measurement accuracy.”

On Page 19, Line 12, we added explanation on the influence of body types on system stability and accuracy and modified the text as “It is worth noting that the distance decoupling model functions as a decoder to directly determine the distance between the implants and the corresponding sensing units. Due to the negligible attenuation of magnetic field strength caused by human tissue^{31,32}, the decoupling model can function effectively across different body types.”

Newly added references:

31. Rubio Ayala, M. *et al.* Spatiotemporal magnetic fields enhance cytosolic Ca²⁺ levels and induce actin polymerization via activation of voltage-gated sodium channels in skeletal muscle cells. *Biomaterials* **163**, 174–184 (2018).

32. Formica, D. & Silvestri, S. Biological effects of exposure to magnetic resonance imaging: an overview. *Biomed. Eng. OnLine* 3, 11 (2004).

Supplementary Fig. 7 | Signal comparison on different device orientation. a. Schematic illustration on device orientation and device response signals (b).

Comment 8: How will the environment influence this device? Will the reading still be accurate when there is a piece of metal near the neck (this can be from a chair, bed, or decoration)?

Our response: We thank the reviewer for the comprehensive comment. Referring to the answers to **Comment 1**, we conducted experiments to study the anti-interference capability of the BioMDA under various external electromagnetic interference, including permanent magnets, metal board, and WIFI router with fixed separation distance to device at 5 cm. The results indicated that the response signals show slight fluctuations in response to different interferences. Moreover, we further reduced the distance between a stainless-steel plane and the device to 1cm, where the response signal presents significant disturbance (**Supplementary Fig. 9**). We added explanations on these results in the text to demonstrate the anti-interference capability of the BioMDA.

Modificatinos: On Page 12, Line 2, we added explanations on the anti-interference capability of the device and modified the text as “In practical applications, the environmental settings and patient conditions are often more complex than controlled lab environments. One of the least controllable variables in clinical settings is the body orientation of patients, which can introduce significant signal interference to the sensing units and decrease measurement/diagnostic accuracy. Therefore, we evaluated the signal stability across a range of device orientations, from -60° to 60° relative to the vertical plane (**Supplementary Fig. 7a**). The response signals in these different orientations showed no significant difference (**Supplementary Fig. 7b**). Additionally, we tested the device’s anti-interference capability against common electromagnetic sources encountered in daily life, including permanent magnets, metal board and WIFI router. A fixed separation distance of 5 cm between the interference sources and the device was maintained to evaluate its anti-interference performance (**Supplementary Fig. 8**). A statical comparison on signal amplitudes revealed only minor fluctuations caused by different external interferences (**Supplementary Fig. 8f**), demonstrating the potential of BioMDA for real world applications. However, further reducing the separation distance between the stainless-steel plane and the device to 1 cm caused significant signal distortion (**Supplementary Fig. 9**), indicating that the user should avoid positioning the device too close (less than 2 cm) to the interference sources during actual use to ensure measurement accuracy.”

Supplementary Fig. 8 | Anti-interference evaluation of the device. Response signals without external interference (a), with horizontal magnet interference (b), vertical magnet interference (c), metal board interference (d), and WIFI router interference (e). f. Static comparison on signal amplitude under different interferences. Square, mean; center line, median; box limits, upper and lower quartiles; whiskers, $1.5 \times$ interquartile range; points, amplitudes in response signals; $n=40$ peak values.

Supplementary Fig. 9 | Anti-interference capability evaluation of the device with interference source distance. a. Schematic illustration on the experimental settings between the interference source and the device. b. Measured response signals under different interference distances.

Response to comments of Referee #2

Summary comment: In this manuscript, Li et al. developed a wearable bio-adhesive metal detector array (BioMDA) for real-time/dynamic postoperative monitoring of spinal implants. This topic is indeed an unaddressed clinical challenge, and the proposed solution using wearable technologies is quite novel and applicable. The authors have provided systematic and solid analyses of proposed technology, including detection principles, optimizations, algorithms, and adhesive materials. There can realize real-time and continuous monitoring of metal implants at non-clinical settings. Thus, it is recommended of the manuscript be published after the following questions:

Our response: We thank the reviewer for these positive comments. We carefully addressed these issues and revised the manuscript accordingly.

Comment 1: Figure 3E demonstrates that the adhesion between skin and skin is stronger than the adhesion between skin and PDMS. Please provide some explanations for this observation. Additionally, is it possible to test the adhesion between PDMS and PDMS to further investigate this issue, as adhesiveness is a key factor in practical applications?

Our response: We thank the reviewer for the valuable comment. We conducted additional experiments to study the interface adhesion performance between PDMS and PDMS to figure out why the adhesion performance in skin-to-skin group is higher than that in skin-to-PDMS group. **Supplementary Fig. 15c** presents the interface adhesion performance comparison in three interface groups, where both interface toughness and shear strength in skin-to-skin group are higher than those in skin-to-PDMS and PDMS-to-PDMS groups. We attributed this difference in adhesion performance to limited hydrogen bonds formed between PDMS and hydrogel. As we know, many amino and carboxyl groups on skin provide sites for hydrogen bonds formation between hydrogel and skin. In comparison, the limited number of amino and carboxyl groups on PDMS surface led to weaker adhesion performance on all PDMS groups. We added a discussion on the interface adhesion performance difference in the manuscript to facilitate better understanding.

Modifications: On Page 14, Line 12, we added a description on the results of adhesion performance in PDMS-to-PDMS group and modified the text as “Interestingly, both the interface toughness and shear strength in the skin-to-skin group were higher than those in the skin-to-PDMS group due to limited hydrogen bonding (**Fig. 3b and 3c**). The interface toughness and shear strength in the PDMS-to-PDMS group presented similar values to those in the skin-to-PDMS group (**Supplementary Fig. 15**), further conforming that the limited hydrogen bond leads to the weaker interface adhesion performance on PDMS. It is worth noting that although the limited hydrogen bonds decrease the interface adhesion performance on PDMS, it is still sufficient for the BioMDA device used in this work.”

Supplementary Fig. 15 | Interface adhesion performance of the bio-adhesive. a. Force variation and shear stress variation (b) to increased separation distance. c. Interface toughness and shear strength comparison of proposed hydrogel in different interface materials.

Comment 2: The authors need to clearly specify the scope of application for this device in a prominent position within the article. For instance, the authors mentioned that current device cannot be applied to non-magnetic implants. Is the device specifically designed for non-magnetic implants only?

Our response: We thank the reviewer for the comment. The current version of BioMDA focused on magnetic implants due to the sensing mechanism of electromagnetic coupling between the sensing units and the implants. Composite materials that involve magnetic materials into the implants or incorporation of an active magnetic excitation may address challenge. We specify the scope of the application and limitations of the current version of BioMDA in the section of Discussion to facilitate better understanding.

Modifications: On Page 22, Line 19, we added a description on the limitations and potential solutions for BioMDA and modified the text as “A number of limitations will need to be addressed in future iterations of the BioMDA system. These are centered on the materials that are able to be

detected (i.e., with the understanding that medical implants have moved away from stainless steel and toward titanium) and the depth of detection (i.e., if such a system is to be applicable in the lumbar spine). Composite materials within the final construct or the incorporation of an active magnetic field excitation may solve the former challenge (e.g., a set cap composed of a nickel alloy vs screw coating as was mentioned above).”

Comment 3: What is the maximum duration the proposed adhesive be maintained on the skin? Will it come off due to sweating/movement/etc.? Can the adhesive be removed/reapplied and for how many cycles before it loses adhesion? If the adhesive is only good for single use, then the technology will be less attractive.

Our response: We thank the reviewer for these comprehensive comments. In practical applications, dehydration of the hydrogel is the key factor in determining the interface stability of the system. To determine the maximum working duration of the bio-adhesive layer, we conducted experiments to study its dehydration performance. In practical application, the bio-adhesive layer functions between the skin and the device to create a seamless interface, which in turn provides an encapsulation to the internal bio-adhesive layer. **Supplementary Fig. 19** presents the comparison on relative mass variation between exposed hydrogel and the hydrogel sealed by skin and device, as schematically illustrated in the inset of **Supplementary Fig. 19a**, for 7 days in 37°C environments. From **Supplementary Fig. 19b**, we can see that over 65% mass of the encapsulated hydrogel was maintained after 7 days exposure in 37°C environments, indicating its normal functionality in interface adhesion.

Additionally, to study if daily activity/sweating will lead to interface adhesion failure due to mechanical deformation or sweat accumulation, we conducted experiments both on an in vivo artificial sweat model and actual wearing conditions on a volunteer. **Supplementary Figs. 24b and 24c** present the optical images of the stable interface connection between the device and artificial skin. While increased surface curvature was caused by increased sweat accumulation in the sweat chamber (**Supplementary Figs. 24b(ii)**), stable interface adhesion was still maintained (**Supplementary Figs. 24c**). Moreover, **Supplementary Figs. 25b** shows the stable interface adhesion was maintained after 30 min intense basketball training.

Furthermore, we tested the adhesion performance between the bio-adhesive and human skin after multiple peel-off cycles. **Supplementary Fig. 17** presents the optical images of the interface adhesion between the bio-adhesive and human skin at peel-off cycle 1 and cycle 5, where it can be seen that after 5 peel-off cycles the interface adhesion performance still kept at satisfied level with noticeable skin deformation generated during the peel-off process (**Supplementary Fig. 17b**). We also adopted a modular design between the bio-adhesive layer and sensing array to facilitate its fast replacement (**Supplementary Fig. 20**).

We added explanations and discussions on these experimental results and modified the text accordingly to facilitate better understanding.

Modifications:

On Page 14, Line 23, we added explanation on the dehydration behavior of the hydrogel and modified the text as “Additionally, the intimate interface between device and skin effectively encapsulated the bio-adhesive layer, preventing interfacial adhesion failure due to water loss (**Supplementary Fig. 19**). We further employed a modular design for the bio-adhesive layer and the sensor array, allowing quick replacement of the adhesive layer while maintaining the functionality of the sensing module (**Supplementary Fig. 20**).”

On Page 16, Line 9, we added discussion on the interface adhesion performance in sweating/movement setting and modified the text as “In practical applications, sweat accumulation poses a significant challenge for achieving robust interface adhesion and maintaining biocompatibility. While commercial double-sided adhesive has proven to be an effective solution for the device-skin interface³⁰, sweat accumulation at the interface can still lead to adhesion failure and skin irritation (**Supplementary Fig. 24**). In contrast, the good water absorption capability of the biocompatible hydrogel prevented sweat accumulation at the interface, thereby avoiding skin irritation and adhesion failure (**Supplementary Figs. 24, 25, Movie 3**).”

On Page 14, Line 19, we added explanation on the interface adhesion performance after multiple peel-off cycles and modified the text as “The interfacial bonding strength also exhibited good robustness, with no evident failure observed after subjecting it to 50 tensile cycles at a 20% tensile strain (**Supplementary Fig. 16**) and more than 5 peel-off cycles from the skin (**Supplementary Fig. 17**).”

Supplementary Fig. 19 | Anti dehydration performance of the bio-adhesive. a. Weight loss comparison on the bio-adhesive under unencapsulated state and encapsulated by device and skin layer for 7 days in 37°C environments. b. Statical comparison on the relative mass variation of two groups of hydrogel during 7 days. Points, mean; error bars, s.d.; n=3 independent samples.

Supplementary Fig. 24 | Interface anti sweat performance comparison between the commercial double-sided adhesive and the bio-adhesive. a. Optical images of the interface adhesion failure with the commercial double-sided adhesive due to sweat accumulation. b. Optical images of the increased surface curvature caused by internal sweat pressure in the bio-adhesive group, proving the seamless interface between the device and the artificial skin. c. Optical images showing the sweat absorption capability of the bio-adhesive under high sweat pressure. Scales bars: 5 mm.

Supplementary Fig. 25 | Interface robustness and biocompatibility comparison between the commercial double-sided adhesive and the bio-adhesive. a. Optical images showing device-skin interface before (i) and after (ii-iv) 30 minutes of intense basketball training. Sweat accumulation and interface delamination were observed after 30 minutes of training. Additionally, skin irritation was also observed due to sweat accumulation on the skin-device interface (iv). b. Optical images showing sweat accumulation and robust device-skin interface in the bio-adhesive group. No sweat irritation was observed in the bio-adhesive group, which can be attributed to the effective sweat absorption reducing sweat accumulation. Scale bars: 1 cm.

Supplementary Fig. 17 | Optical images of the bio-adhesive after multi peel-off cycles. The skin deformation indicates the bio-adhesive maintained satisfied interface adhesion strength after 5 peel-off cycles. Scale bars: 1 cm.

Supplementary Fig. 20 | Optical images of the modular design of the bio-adhesive layer. Scale bars: 1 cm.

Comment 4: Following the previous question, there is no data of the device during actual wearing by a human, other than a few photographs in Fig. 1. The authors are suggested to do more mechanical and electrical evaluations on real human testing subjects and evaluate the device's long-term wearability, sweat-resisting ability, signal resistance to human motion artifacts etc. during actual wearing.

Our response: We thank the reviewer for these comprehensive comments. Referring to the answers to **Comment 3**, we conducted experiments on in vivo artificial sweat model and human subject to evaluate the long-term stability and sweat-resisting capability of the device. The results indicated that the bio-adhesive maintained a robust interface under both high sweat pressure (in vivo artificial sweat model) and intense basketball training (user study). Moreover, to evaluate the anti-interference capability of the device, we conducted experiments to evaluate signals stability under a set of device orientation and different external electromagnetic interference sources. **Supplementary Fig. 7** and **Fig. 8** present that stable signals were generated under different device orientations and external interference, indicating the good anti-interference capability of the device.

Modifications:

On Page 16, Line 9, we added an explanation on the comparison on commercial double-sided adhesive and bio-adhesive and modified the text as “In practical applications, sweat accumulation poses a significant challenge for achieving robust interface adhesion and maintaining biocompatibility. While commercial double-sided adhesive has proven to be an effective solution for the device-skin interface³⁰, sweat accumulation at the interface can still lead to adhesion failure and skin irritation (**Supplementary Fig. 24**). In contrast, the good water absorption capability of the biocompatible hydrogel prevented sweat accumulation at the interface, thereby avoiding skin irritation and adhesion failure (**Supplementary Figs. 24, 25, Movie 3**).”

On Page 12, Line 2, we added explanations on the anti-interference capability of the device and modified the text as “In practical applications, the environmental settings and patient conditions

are often more complex than controlled lab environments. One of the least controllable variables in clinical settings is the body orientation of patients, which can introduce significant signal interference to the sensing units and decrease measurement/diagnostic accuracy. Therefore, we evaluated the signal stability across a range of device orientations, from -60° to 60° relative to the vertical plane (**Supplementary Fig. 7a**). The response signals in these different orientations showed no significant difference (**Supplementary Fig. 7b**). Additionally, we tested the device's anti-interference capability against common electromagnetic sources encountered in daily life, including permanent magnets, metal board and WIFI router. A fixed separation distance of 5 cm between the interference sources and the device was maintained to evaluate its anti-interference performance (**Supplementary Fig. 8**). A statical comparison on signal amplitudes revealed only minor fluctuations caused by different external interferences (**Supplementary Fig. 8f**), demonstrating the potential of BioMDA for real world applications. However, further reducing the separation distance between the stainless-steel plane and the device to 1 cm caused significant signal distortion (**Supplementary Fig. 9**), indicating that the user should avoid positioning the device too close (less than 2 cm) to the interference sources during actual use to ensure measurement accuracy.”

Supplementary Fig. 7 | Signal comparison on different device orientation. a. Schematic illustration on device orientation and device response signals (b).

Supplementary Fig. 8 | Anti-interference evaluation of the device. Response signals without external interference (a), with horizontal magnet interference (b), vertical magnet interference (c), metal board interference (d), and WIFI router interference (e). f. Static comparison on signal amplitude under different interferences. Square, mean; center line, median; box limits, upper and lower quartiles; whiskers, $1.5 \times$ interquartile range; points, amplitudes in response signals; $n=40$ peak values.

Comment 5: The position and data of Fig. 2I for "holding" overlap.

Our response: We thank the reviewer for the comment. We modified Fig. 2I accordingly to avoid data curve and heading overlap.

Modifications: On Page 37, Line 1, we modified Fig. 2 as follows.

Fig. 2 | Structural parameters optimization and characterization of the single sensing unit. a. Schematic illustration of the components comprising a single sensing unit. b. Force state analysis of the permanent magnet in response to external attractive force and structural parameters that determine the sensing capability and stability of the unit. Schematic illustration of the platform (c) utilized to study the force changes with increased deviating distance (d) with nickel coating thickness ranging from 0 to 200 nm. e. Finite element analysis results showing the strain distribution of the supporting PET film with thickness of 50 μm and 250 μm in response to the external attractive force of 100 mN. f. Measured response signal amplitude (f) and maximum detection distance (g) variation with a set of film thicknesses ranging from 50 μm to 250 μm . h. Measured response signal amplitude with central angle of the PET film ranging from 30° to 90°. i. Measured response signal showing the cyclic variation during implant approaching, holding, and deviating process. j. Fast Fourier Transform (FFT) of the response signal in (i) showing the movement frequency of the implant. k. Response signals in over 4000 approaching and deviating cycles showing the stability of the single sensing unit.

Comment 6: The font of the y-axis in Fig. 2D should be consistent with the other figures.

Our response: We thank the reviewer for the comment. We standardized all the fonts and font sizes in the figure.

Modifications: On Page 37, Line 1, we modified Fig. 2 as follows.

Fig. 2 | Structural parameters optimization and characterization of the single sensing unit. a. Schematic illustration of the components comprising a single sensing unit. b. Force state analysis of the permanent magnet in response to external attractive force and structural parameters that determine the sensing capability and stability of the unit. Schematic illustration of the platform (c) utilized to study the force changes with increased deviating distance (d) with nickel coating thickness ranging from 0 to 200 nm. e. Finite element analysis results showing the strain distribution of the supporting PET film with thickness of 50 μm and 250 μm in response to the external attractive force of 100 mN. f. Measured response signal amplitude (f) and maximum detection distance (g) variation with a set of film thicknesses ranging from 50 μm to 250 μm . h. Measured response signal amplitude with central angle of the PET film ranging from 30° to 90°. i. Measured response signal showing the cyclic variation during implant approaching, holding, and deviating process. j. Fast Fourier Transform (FFT) of the response signal in (i) showing the

movement frequency of the implant. k. Response signals in over 4000 approaching and deviating cycles showing the stability of the single sensing unit.

Comment 7: The authors need to double-check the mathematical symbols in lines 280-308. Some symbols appear to have inconsistent font sizes and appear compressed, such as Line 284 and Line 286.

Our response: We thank the reviewer for the comment. We carefully checked and addressed all the formats issues with mathematical symbols.

Modifications:

On Page 17, Line 17, we uniformed the font size of Equations 1-3 and modified the text as “

$$B = \frac{\mu_0 M}{2} \left[\frac{z}{\sqrt{z^2 + a^2}} - \frac{z-h}{\sqrt{(z-h)^2 + a^2}} \right] \quad (1)$$

$$\varepsilon_0(z) = -\frac{d}{dt} \int_{\Sigma} B dA = -\frac{\mu_0 N M A a^2}{2} [(z^2 + a^2)^{-1.5} - ((z-h)^2 + a^2)^{-1.5}] \frac{dz}{dt} \quad (2)$$

$$\varepsilon(z) = e^{-\gamma z} \varepsilon_0(z) = e^{-\gamma z} \xi_B(z) \dot{z} \quad (3)$$

”

On Page 18, Line 14, we uniformed the font size of Equations 4-6 and modified the text as “

$$SF(z, \dot{z}) = sf_1(z) + sf_2(z)(\dot{z} - (kz + b)) \quad (4)$$

$$sf_*(z) = a_1 e^{-z} + a_0 \quad (5)$$

$$\hat{\varepsilon}(z_m) \approx SF(z_i, \dot{z}_i) \xi_B(\max(z_i, z_0)) \dot{z}_i \quad (6)$$

”

Comment 8: This BioMDA device used self-adhesive hydrogel. What is the maximum working time of this device, and will long-term attachment to human skin cause inflammation or discomfort?

Our response: We thank the reviewer for the valuable comment. Referring our response to **Comment 3**, the hydrogel can maintain over 65% relative mass after storing in 37°C environments for 7 days, indicating its long-term stability and wearability. Moreover, we conducted experiments on two volunteers to study if skin irritation/discomfort caused by the hydrogel after long-term attachment to human skin. **Supplementary Fig. 11** shows the skin irritation results of three types of materials on the forearms of two volunteers, where no skin irritation or inflammation were observed in all three groups after continuous 12 h covering. We added explanation on these results in the text to prove the biocompatibility of the proposed bio-adhesive.

Modifications: On Page 13, Line 8, we added explanation on the skin irritation results and modified the text as “The adhesive hydrogel consists of a poly (acrylic acid) (PAA) network crosslinked with biodegradable gelatin methacrylate, along with a biodegradable gelatin network (Fig. 3a), showing strong adhesion to diverse materials (Supplementary Fig. 10) and superior biocompatibility, without skin irritation caused on 2 volunteers after 12 h continuous covering (Supplementary Fig. 11).”

Supplementary Fig. 11 | Biocompatibility of the developed bio-adhesive. a-b. Optical images of two volunteer’s forearms taken before and after being covered 12 hours with hydrogel, PDMS, and commercial double-sided adhesive.

Comment 9: How to assess the comfort during the wear period of BioMDA.

Our response: We thank the reviewer for the comment. We assess the user comfort from the weight, flexibility of the device, and the skin irritation. In the BioMDA system, we utilized ultrathin, flexible design to the interconnect traces, conducted lightweight optimization to the sensing units, developed biocompatible interface adhesion layer to maximum user comfort mechanically and biologically. Specifically, the mechanical design facilitates a lightweight (40.3 g) and flexible sensing array (Supplementary Figs. 2 and 3). Moreover, the biocompatible interface adhesive layer facilitates user friendly interface without skin irritation caused after long-term use (Supplementary Fig. 11). We added this description in the manuscript to better present the user comfort of the BioMDA system.

Modifications:

On Page 7, Line 6, we added the weight of the BioMDA and modified the text as “The robust interface connection, flexible designs, and integration strategies employed in the BioMDA resulted in and ultralight system (weighting only 40.3 g) with exceptional flexibility and interface stability of the sensor array, facilitating intimate contact between the BioMDA and patient skin as well as long-term comfortability (Fig. 1c, Supplementary Fig. 3).”

On Page 13, Line 8, we added the description on biocompatibility and modified the text as “The adhesive hydrogel consists of a poly (acrylic acid) (PAA) network crosslinked with biodegradable gelatin methacrylate, along with a biodegradable gelatin network (Fig. 3a), showing strong adhesion to diverse materials (Supplementary Fig. 10) and superior biocompatibility, without skin irritation caused on 2 volunteers after 12 h continuous covering (Supplementary Fig. 11).”

Supplementary Fig. 2 | Optical images of the flexible electrode of the BioMDA. a. Optical images of the electrode and enlarged view of the integrated connection port. Optical images of the flexible electrode undergoing twisting deformation (b) and stretching deformation (c). Scale bars: 1 cm.

Supplementary Fig 3 | Optical images of the single sensing unit and assembled sensing units into a 4×4 sensing array. a. Optical image of the single sensing unit. b. Assembled sensing array. c. Assembled sensing array in bending state. Scale bars: 5 mm.

Supplementary Fig. 11 | Biocompatibility of the developed bio-adhesive. a-b. Optical images of two volunteer's forearms taken before and after being covered 12 hours with hydrogel, PDMS, and commercial double-sided adhesive.

Comment 10: Isopropanol (toxic) was used in the synthesis process of the material. How should one determine that the isopropanol or other harmful substances used during the material synthesis will not be harmful to the human body?

Our response: We thank the reviewer for the valuable comment. During the preparation of flexible interconnects and amine grafted PDMS, isopropanol was used to clean the substrate. While the isopropanol was isolated by the bio-adhesive hydrogel to human skin, residual traces on the surface may still cause skin irritation. To alleviate health concerns, we involved two cleaning steps by ethanol and DI water subsequently to remove isopropanol thoroughly.

Modifications: On Page 26, Line 11, we involved two cleaning steps to the fabrication of amino grafted PDMS and modified the text as “Then, the activated stretchable electrode was immersed in 1% (w/w) APTES solution (1% (w/w) APTES in 50% ethanol) and incubated for 5 hours at room temperature before cleaning it with isopropyl alcohol, ethanol, and DI water.”

REVIEWERS' COMMENTS

Reviewer #1 (Remarks to the Author):

The authors have fully addressed my previous comments.

Reviewer #2 (Remarks to the Author):

The authors have addressed all the issues about the questions, and recommendation for publication as it is now.